# Standardized and reproducible measurement of decision-making in mice

The International Brain Laboratory*, Valeria Aguillon-Rodriguez[1], Dora Angelaki[2], Hannah Bayer[3], Niccolo Bonacchi[4], Matteo Carandini[5], Fanny Cazettes[4], Gaelle Chapuis[6], Anne K Churchland[1†], Yang Dan[7], Eric Dewitt[4], Mayo Faulkner[6], Hamish Forrest[5], Laura Haetzel[8], Michael Häusser[6], Sonja B Hofer[9], Fei Hu[7], Anup Khanal[1†], Christopher Krasniak[1,10], Ines Laranjeira[4], Zachary F Mainen[4], Guido Meijer[4], Nathaniel J Miska[9], Thomas D Mrsic-Flogel[9], Masayoshi Murakami[4‡], Jean-Paul Noel[2], Alejandro Pan-Vazquez[8], Cyrille Rossant[11], Joshua Sanders[12], Karolina Socha[5], Rebecca Terry[11], Anne E Urai[1,13], Hernando Vergara[9], Miles Wells[11], Christian J Wilson[2], Ilana B Witten[8], Lauren E Wool[11], Anthony M Zador[1]

[1]Cold Spring Harbor Laboratory, New York, United States; [2]Center for Neural Science, New York University, New York, United States; [3]Zuckerman Institute, Columbia University, New York, United States; [4]Champalimaud Centre for the Unknown, Lisbon, Portugal; [5]UCL Institute of Ophthalmology, University College London, London, United Kingdom; [6]Wolfson Institute for Biomedical Research, University College London, London, United Kingdom; [7]Department of Molecular and Cell Biology, University of California, Berkeley, Berkeley, United States; [8]Princeton Neuroscience Institute, Princeton University, Princeton, United States; [9]Sainsbury-Wellcome Centre for Neural Circuits and Behaviour, University College London, London, United Kingdom; [10]Watson School of Biological Sciences, New York, United States; [11]UCL Queen Square Institute of Neurology, University College London, London, United Kingdom; [12]Sanworks LLC, New York, United States; [13]Cognitive Psychology Unit, Leiden University, Leiden, Netherlands

*For correspondence:
info+behavior@
internationalbrainlab.org

Present address: †David Geffen School of Medicine, University of California, Los Angeles, Los Angeles, United States; ‡Department of Neurophysiology, University of Yamanashi, Kōfu, Japan

**Abstract** Progress in science requires standardized assays whose results can be readily shared, compared, and reproduced across laboratories. Reproducibility, however, has been a concern in neuroscience, particularly for measurements of mouse behavior. Here, we show that a standardized task to probe decision-making in mice produces reproducible results across multiple laboratories. We adopted a task for head-fixed mice that assays perceptual and value-based decision making, and we standardized training protocol and experimental hardware, software, and procedures. We trained 140 mice across seven laboratories in three countries, and we collected 5 million mouse choices into a publicly available database. Learning speed was variable across mice and laboratories, but once training was complete there were no significant differences in behavior across laboratories. Mice in different laboratories adopted similar reliance on visual stimuli, on past successes and failures, and on estimates of stimulus prior probability to guide their choices. These results reveal that a complex mouse behavior can be reproduced across multiple laboratories. They establish a standard for reproducible rodent behavior, and provide an unprecedented dataset and open-access tools to study decision-making in mice. More generally, they indicate a path toward achieving reproducibility in neuroscience through collaborative open-science approaches.

**eLife digest** In science, it is of vital importance that multiple studies corroborate the same result. Researchers therefore need to know all the details of previous experiments in order to implement the procedures as exactly as possible. However, this is becoming a major problem in neuroscience, as animal studies of behavior have proven to be hard to reproduce, and most experiments are never replicated by other laboratories.

Mice are increasingly being used to study the neural mechanisms of decision making, taking advantage of the genetic, imaging and physiological tools that are available for mouse brains. Yet, the lack of standardized behavioral assays is leading to inconsistent results between laboratories. This makes it challenging to carry out large-scale collaborations which have led to massive breakthroughs in other fields such as physics and genetics.

To help make these studies more reproducible, the International Brain Laboratory (a collaborative research group) et al. developed a standardized approach for investigating decision making in mice that incorporates every step of the process; from the training protocol to the software used to analyze the data. In the experiment, mice were shown images with different contrast and had to indicate, using a steering wheel, whether it appeared on their right or left. The mice then received a drop of sugar water for every correction decision. When the image contrast was high, mice could rely on their vision. However, when the image contrast was very low or zero, they needed to consider the information of previous trials and choose the side that had recently appeared more frequently.

This method was used to train 140 mice in seven laboratories from three different countries. The results showed that learning speed was different across mice and laboratories, but once training was complete the mice behaved consistently, relying on visual stimuli or experiences to guide their choices in a similar way.

These results show that complex behaviors in mice can be reproduced across multiple laboratories, providing an unprecedented dataset and open-access tools for studying decision making. This work could serve as a foundation for other groups, paving the way to a more collaborative approach in the field of neuroscience that could help to tackle complex research challenges.

## Introduction

Progress in science depends on reproducibility and thus requires standardized assays whose methods and results can be readily shared, compared, and reproduced across laboratories (*Baker, 2016*; *Ioannidis, 2005*). Such assays are common in fields such as astronomy (*Fish et al., 2016*; *Abdalla et al., 2018*), physics (*CERN Education, Communications and Outreach Group, 2018*), genetics (*Dickinson et al., 2016*), and medicine (*Bycroft et al., 2018*), and perhaps rarer in fields such as sociology (*Camerer et al., 2018*) and psychology (*Forscher et al., 2020*; *Frank et al., 2017*; *Makel et al., 2012*). They are also rare in neuroscience, a field that faces a reproducibility crisis (*Baker, 2016*; *Botvinik-Nezer et al., 2020*; *Button et al., 2013*).

Reproducibility has been a particular concern for measurements of mouse behavior (*Kafkafi et al., 2018*). Although the methods can be generally reproduced across laboratories, the results can be surprisingly different ('methods reproducibility' vs. 'results reproducibility', *Goodman et al., 2016*). Even seemingly simple assays of responses to pain or stress can be swayed by extraneous factors (*Chesler et al., 2002*; *Crabbe et al., 1999*) such as the sex of the experimenter (*Sorge et al., 2014*). Behavioral assays can be difficult to reproduce across laboratories even when they share a similar apparatus (*Chesler et al., 2002*; *Crabbe et al., 1999*; *Sorge et al., 2014*). This difficulty is not simply due to genetic variation: behavioral variability is as large in inbred mice as in outbred mice (*Tuttle et al., 2018*).

Difficulties in reproducing mouse behavior across laboratories would hinder the increasing number of studies that investigate decision making in mice. Physiological studies of decision making are increasingly carried out in mice to access the unrivaled arsenal of genetic, imaging, and physiological tools available for mouse brains (*Carandini and Churchland, 2013*; *Glickfeld et al., 2014*; *O'Connor et al., 2009*). Our collaboration, *International Brain Laboratory, 2017*, aims to leverage

these approaches by exploring the neural basis of the same mouse behavior in multiple laboratories. It is thus crucial for this endeavor that the relevant behavioral assays be reproducible both in methods and in results.

Studying decision-making requires a task that places specific sensory, cognitive, and motor demands over hundreds of trials, affording strong constraints to behavior. The task should be complex enough to expose the neural computations that support decision-making but simple enough for mice to learn, and easily extendable to study further aspects of perception and cognition. Moreover, it can be invaluable to have ready access to the brain for neural recordings and manipulations, a consideration that favors tasks that involve head fixation.

To meet these criteria, we adopted a modified version of the classical 'two-alternative forced-choice' perceptual detection task. In the classical task, the subject indicates the position of a stimulus that can be in one of two positions with equal probability (e.g. *Carandini and Churchland, 2013*; *Tanner and Swets, 1954*). In the modified version, the probability of the stimulus being in one position changes over time (*Terman and Terman, 1972*). This change in probability may affect sensory decisions by directing spatial attention (*Cohen and Maunsell, 2009*; *Liston and Stone, 2008*) and by biasing the decision process (*Hanks et al., 2011*). It modifies the expected value of the choices, echoing changes in reward probability or size, which affect perceptual choices (e.g. *Feng et al., 2009*; *Whiteley and Sahani, 2008*) and drive value-based choices (*Corrado et al., 2005*; *Fan et al., 2018*; *Herrnstein, 1961*; *Lau and Glimcher, 2005*; *Miller et al., 2019*).

In the task, mice detect the presence of a visual grating to their left or right, and report the perceived location with a simple movement: by turning a steering wheel (*Burgess et al., 2017*). The task difficulty is controlled by varying the contrast across trials. The reward for a correct response is a drop of sugar water that is not contingent on licking the spout. The probability of stimulus appearance at the two locations is asymmetric and changes across blocks of trials. Mice thus make decisions by using both their vision and their recent experience. When the visual stimulus is evident (contrast is high), they should mostly use vision, and when the visual stimulus is ambiguous (contrast is low or zero), they should consider prior information (*Whiteley and Sahani, 2008*) and choose the side that has recently been more likely.

We here present results from a large cohort of mice trained in the task, demonstrating reproducible methods and reproducible results across laboratories. In all laboratories, most mice learned the task, although often at a different pace. After learning they performed the task in a comparable manner, and with no significant difference across laboratories. Mice in different laboratories adopted a comparable reliance on visual stimuli, on past successes and failures, and on estimates of stimulus prior probability.

To facilitate reuse and reproducibility, we adopt an open science approach: we describe and share the hardware and software components and the experimental protocols. It is increasingly recognized that data and techniques should be made fully available to the broader community (*Beraldo et al., 2019*; *Charles et al., 2020*; *Forscher et al., 2020*; *Koscielny et al., 2014*; *Poldrack and Gorgolewski, 2014*; *de Vries et al., 2020*). Following this approach, we established an open-access data architecture pipeline (*Bonacchi et al., 2020*) and use it to release the >5 million mouse choices at data.internationalbrainlab.org. These results reveal that a complex mouse behavior can be successfully reproduced across laboratories, enabling collaborative studies of brain function in behaving mice.

## Results

To train mice consistently within and across laboratories we developed a standardized training pipeline (*Figure 1a*). First, we performed surgery to implant a headbar for head-fixation (*IBL Protocol for headbar implant surgery in mice* [*The International Brain Laboratory, 2020a*]). During the subsequent recovery period, we handled the mice and weighed them daily. Following recovery, we put mice on water control and habituated them to the experimental setup (*IBL Protocol for mice training* [*The International Brain Laboratory, 2020b*]). Throughout these steps, we checked for adverse effects such as developing a cataract during surgery or pain after surgery or substantial weight loss following water control (four mice excluded out of 210) (*Guo et al., 2014*).

Mice were then trained in two stages: first they learned a *basic task* (*Burgess et al., 2017*), where the probability of a stimulus appearing on the left or the right was equal (50:50), and then they

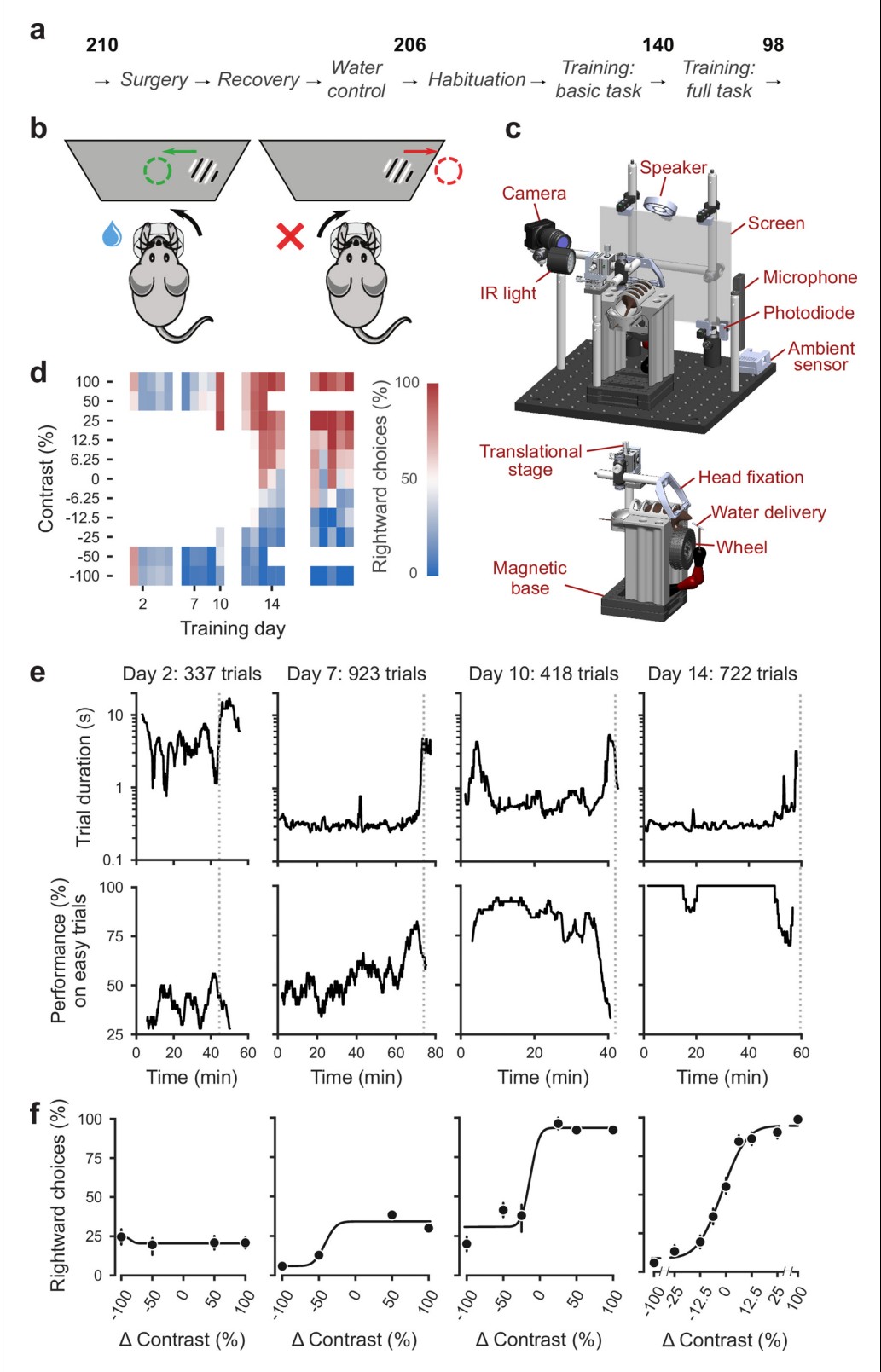

**Figure 1.** Standardized pipeline and apparatus, and training progression in the basic task. (**a**) The pipeline for mouse surgeries and training. The number of animals at each stage of the pipeline is shown in bold. (**b**) Schematic of the task, showing the steering wheel and the visual stimulus moving to the center of the screen vs. the opposite direction, with resulting reward vs. timeout. (**c**) CAD model of the behavioral apparatus. *Top*: the entire apparatus, *Figure 1 continued on next page*

*Figure 1 continued*

showing the back of the mouse. The screen is shown as transparent for illustration purposes. *Bottom*: side view of the detachable mouse holder, showing the steering wheel and water spout. A 3D rendered video of the CAD model can be found here. (**d**) Performance of an example mouse (KS014, from Lab 1) throughout training. Squares indicate choice performance for a given stimulus on a given day. Color indicates the percentage of right (*red*) and left (*blue*) choices. Empty squares indicate stimuli that were not presented. Negative contrasts denote stimuli on the left, positive contrasts denote stimuli on the right. (**e**) Example sessions from the same mouse. *Vertical lines* indicate when the mouse reached the session-ending criteria based on trial duration (*top*) and accuracy on high-contrast (>=50%) trials (*bottom*) averaged over a rolling window of 10 trials (*Figure 1—figure supplement 1*). (**f**) Psychometric curves for those sessions, showing the fraction of trials in which the stimulus on the right was chosen (rightward choices) as a function of stimulus position and contrast (difference between right and left, i.e. positive for right stimuli, negative for left stimuli). Circles show the mean and error bars show ±68% confidence intervals. The training history of this mouse can be explored at this interactive web page.

The online version of this article includes the following figure supplement(s) for figure 1:

**Figure supplement 1.** Task trial structure.
**Figure supplement 2.** Distribution of within-session disengagement criteria.

learned the *full task*, where the probability of stimuli appearing on the left vs. right switched in blocks of trials between 20:80 and 80:20. Out of 206 mice that started training, 140 achieved proficiency in the basic task, and 98 in the full task (see *Appendix 1—table 1d* for training progression and proficiency criteria). The basic task is purely perceptual: the only informative cue is the visual stimulus. The full task instead invites integration of perception with recent experience: when stimuli are ambiguous it is best to choose the more likely option.

To facilitate reproducibility, we standardized multiple variables, measured multiple other variables, and shared the behavioral data in a database that we inspected regularly. By providing standards and guidelines, we sought to control variables such as mouse strain and provider, age range, weight range, water access, food protein, and fat. We did not attempt to standardize other variables such as light-dark cycle, temperature, humidity, and environmental sound, but we documented and measured them regularly (*Appendix 1—table 1a*; *Voelkl et al., 2020*). Data were shared and processed across the collaboration according to a standardized web-based pipeline (*Bonacchi et al., 2020*). This pipeline included a colony management database that stored data about each session and mouse (e.g. session start time, animal weight, etc.), a centralized data repository for files generated in the task (e.g. behavioral responses and compressed video and audio files), and a platform which provided automated analyses and daily visualizations (data.internationalbrainlab.org) (*Yatsenko et al., 2018*).

Mice were trained in a standardized setup involving a steering wheel placed in front of a screen (*Figure 1b,c*). The visual stimulus, a grating, appeared at variable contrast on the left or right half of the screen. The stimulus position was coupled with movements of the response wheel, and mice indicated their choices by turning the wheel left or right to bring the grating to the center of the screen (*Burgess et al., 2017*). Trials began after the mouse held the wheel still for 0.4–0.7 s and were announced by an auditory 'go cue'. Correct decisions were rewarded with sweetened water (10% sucrose solution), whereas incorrect decisions were indicated by a noise burst and were followed by a longer inter-trial interval (2 s) (*Guo et al., 2014*, *Figure 1—figure supplement 1*). The experimental setups included systems for head-fixation, visual and auditory stimuli presentation, and recording of video and audio (*Figure 1c*). These were standardized and based on open-source hardware and software (*IBL protocol for setting up the behavioral training rig* [*The International Brain Laboratory, 2021a*]).

## Training progression in the basic task

We begin by describing training in the basic task, where stimuli on the left vs. right appeared with equal probability (*Burgess et al., 2017*). This version of the task is purely visual, in that no other information can be used to increase expected reward.

The training proceeded in automated steps, following predefined criteria (*Figure 1d*, *IBL Protocol for mice training*). Initially, mice experienced only easy trials with highly visible stimuli (100% and 50% contrast). As performance improved, the stimulus set progressively grew to include contrasts of

25%, 12%, 6%, and finally 0% (*Figure 1d*, *Appendix 1—table 1b-c*). Stimuli with contrast >0 could appear on the left or the right and thus appeared twice more often than stimuli with 0% contrast. For a typical mouse (*Figure 1d*), according to this automated schedule, stimuli with 25% contrast were introduced in training day 10, 12% contrast in Day 12, and the remaining contrasts in Day 13. On this day the 50% contrast trials were dropped, to increase the proportion of low-contrast trials. To reduce response biases, incorrect responses on easy trials (high contrast) were more likely to be followed by a 'repeat trial' with the same stimulus contrast and location.

On each training day, mice were typically trained in a single uninterrupted session, whose duration depended on performance (*Figure 1e*). Sessions lasted at most 90 min and ended according to a criterion based on number of trials, total duration, and response times (*Figure 1—figure supplement 2*). For instance, for the example mouse, session on Day 2 ended when 45 min elapsed with <400 trials; sessions on Days 7, 10, and 14 ended when trial duration increased to five times above baseline (*Figure 1e*). The criterion to end a session was not mandatory: some sessions were ended earlier (5,316/10,903 sessions) and others ended later (5,587/10,903 sessions). The latter sessions typically continued for an extra 19 ± 16 trials (median ±m.a.d., 5,587 sessions) longer.

To encourage vigorous motor responses and to increase the number of trials, the automated protocol increased the motor demands and reduced reward volume over time. At the beginning of training, the wheel gain was high (8 deg/mm), making the stimuli highly responsive to small wheel movements, and rewards were large (3 µL). Once a session had >200 complete trials, the wheel gain was halved to 4 deg/mm and the reward was progressively decreased to 1.5 µL (*Appendix 1—table 1b-c*).

Mouse performance gradually improved until it provided high-quality visual psychometric curves (*Figure 1f*). At first, performance hovered around or below 50%. It could be even below 50% because of no-response trials (which were labeled as incorrect trials in our analyses), systematic response biases, and the bias-correcting procedure that tended to repeat trials following errors on easy trials (at high contrast). Performance then typically increased over days to weeks until mice made only rare mistakes (lapses) on easy trials (e.g. *Figure 1f*, Day 14).

Animals were considered to have reached proficiency in the basic task when they had been introduced to all contrast levels, and had met predefined performance criteria based on the parameters of the psychometric curves fitted to the data (Materials and methods). These parameters and the associated criteria were as follows: (1) response bias (estimated from the horizontal position of the psychometric curve): maximum absolute value of <16% contrast; (2) contrast sensitivity (estimated from the slope of the psychometric curve): maximum threshold (1/sensitivity) of 19% contrast; (3) lapse rates (estimated from the asymptotes of the psychometric curve): maximum of 0.2 for their sum. These criteria had to be fulfilled on three consecutive training sessions (*Figure 1* , *Appendix 1—table d*). The training procedure, performance criteria, and psychometric parameters are described in detail in IBL Protocol for mice training (*The International Brain Laboratory, 2020b*).

## Training succeeded but with different rates across mice and laboratories

The training procedures succeeded in all laboratories, but the duration of training varied across mice and laboratories (*Figure 2*). There was substantial variation among mice, with the fastest learner achieving basic task proficiency in 3 days, and the slowest after 59 days (*Figure 2a*). Estimates of contrast threshold did not vary much during training (*Figure 2b*), with an interquartile range that slightly decreased from 12–29% during the first 10 training days, and 11–23% thereafter. Likewise, bias did not vary much, with an interquartile range of −5.2% to 7.1% during the first 10 days to −5.6% to 6.5% thereafter (*Figure 2c*). These effects were similar across laboratories (*Figure 2e,f*). The average training took 18.4 ± 13.0 days (s.d., n = 140, *Figure 2g*). The number of days needed to achieve basic task proficiency was different across laboratories (*Figure 2g*, p<0.001, Kruskal-Wallis nonparametric test followed by a post-hoc Dunn's multiple comparisons test). Some labs had homogeneous learning rates (e.g. Lab two within-lab interquartile range of 8 days), while other labs had larger variability (e.g. Lab six interquartile range of 22 days).

These differences in learning rates across laboratories could not be explained by differences in number of trials per session. To account for these possible differences, we measured training proficiency across trials rather than days (*Figure 2—figure supplement 1*). For mice that learned the task, the average training took 10.8 ± 8.6 thousands of trials (s.d., n = 140), similar to the 13

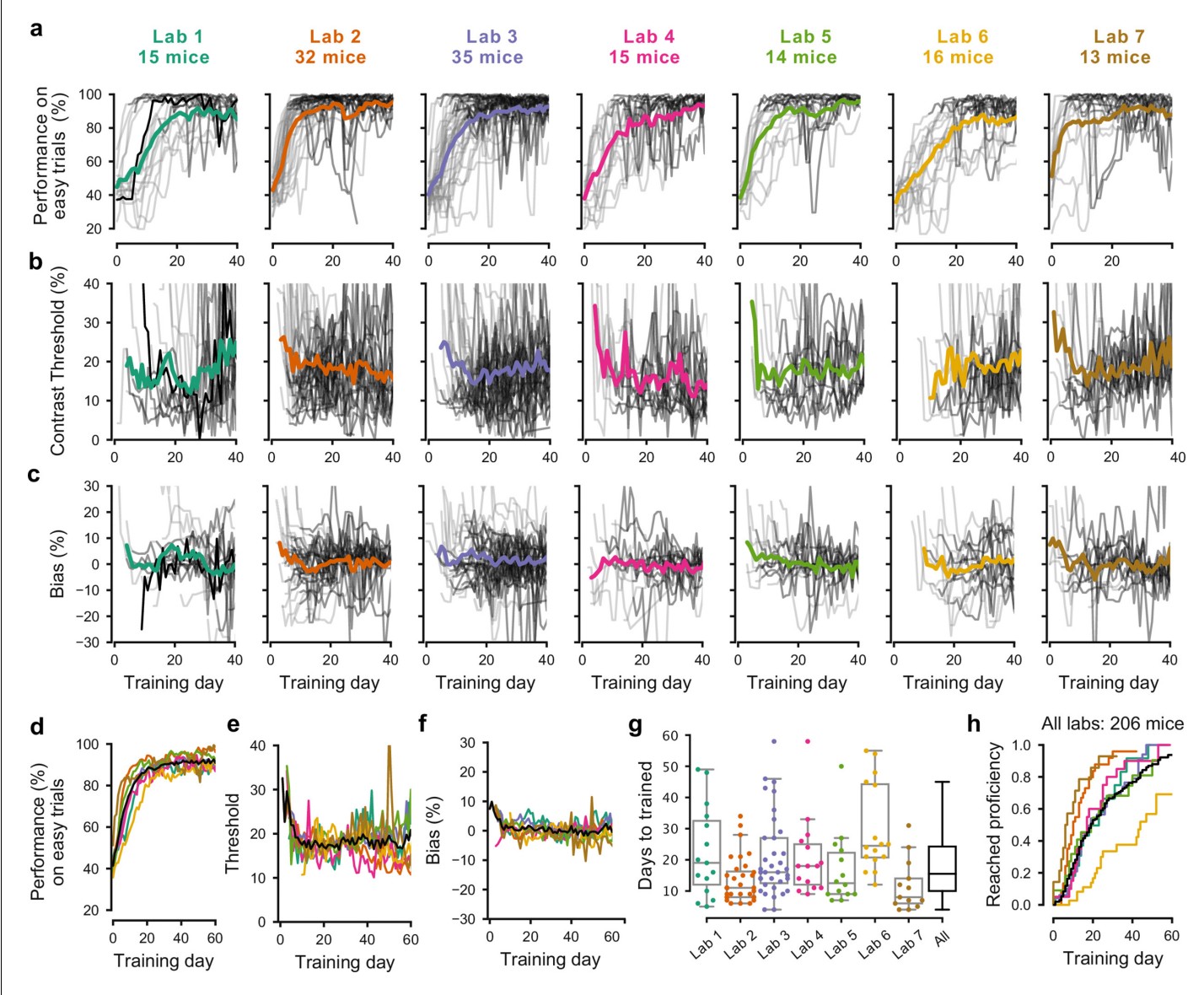

**Figure 2.** Learning rates differed across mice and laboratories. (**a**) Performance for each mouse and laboratory throughout training. Performance was measured on easy trials (50% and 100% contrast). Each panel represents a different lab, and each thin curve represents a mouse. The transition from *light gray* to *dark gray* indicates when each mouse achieved proficiency in the basic task. *Black*, performance for example mouse in *Figure 1*. Thick colored lines show the lab average. Curves stop at day 40, when the automated training procedure suggests that mice be dropped from the study if they have not learned. (**b**) Same, for contrast threshold, calculated starting from the first session with a 12% contrast (i.e. first session with six or more different trial types), to ensure accurate psychometric curve fitting. Thick colored lines show the lab average from the moment there were three or more datapoints for a given training day. (**c**) Same, for choice bias. (**d-f**) Average performance, contrast threshold and choice bias of each laboratory across training days. *Black curve* denotes average across mice and laboratories (**g**) Training times for each mouse compared to the distribution across all laboratories (*black*). *Boxplots* show median and quartiles. (**h**) Cumulative proportion of mice to have reached proficiency as a function of training day (Kaplan-Meier estimate). *Black curve* denotes average across mice and laboratories. Data in (**a-g**) is for mice that reached proficiency (n = 140). Data in **h** is for all mice that started training (n = 206).

The online version of this article includes the following figure supplement(s) for figure 2:

**Figure supplement 1.** Learning rates measured by trial numbers.

**Figure supplement 2.** Performance variability within and across laboratories decreases with training.

thousand trials of the example mouse from Lab 1 (*Figure 2a*, black). The fastest learner met training criteria in one thousand trials, the slowest 43 thousand trials. To reach 80% on easy trials took on average ~7.4 thousand trials.

Variability in learning rates was also not due to systematic differences in choice bias or visual sensitivity (*Figure 2b–c*). Mice across laboratories were not systematically biased towards a particular side and the absolute bias was similar across laboratories (9.8 ± 12.1 [s.d.] on average). Likewise, measures of contrast threshold stabilized after the first ~10 sessions, which was on average 17.8 ± 11.7% on average across institutions.

The variability in performance across mice decreased as training progressed, but it did not disappear (*Figure 2—figure supplement 2*). Variability in performance was larger in the middle of training than towards the end. For example, between training days 15 and 40, when the average performance on easy trials of the mice increased from 80.7% to 91.1%, the variation across mice (s. d.) of performance on easy trials decreased from 19.1% to 10.1%.

To some extent, a mouse's performance in the first five sessions predicted how long it would take the mouse to become proficient. A Random Forests decoder applied to change in performance (% correct with easy, high-contrast stimuli) in the first five sessions was able to predict whether a mouse would end up in the bottom quartile of learning speed (the slowest learners) with accuracy of 53% (where chance is 25%). Conversely, the chance of misclassifying a fast-learning, top quartile mouse as a slow-learning, bottom quartile mouse, was only 7%.

Overall, our procedures succeeded in training tens of mice in each laboratory (*Figure 2h*). Indeed, there was a 80% probability that mice would learn the task within the 40 days that were usually allotted, and when mice were trained for a longer duration, the success rate rose even further (*Figure 2h*). There was, however, variability in learning rates across mice and laboratories. This variability is intriguing and would present a challenge for projects that aim to study learning. We next ask whether the behavior of trained mice was consistent and reproducible across laboratories.

## Performance in the basic task was indistinguishable across laboratories

Once mice achieved basic task proficiency, multiple measures of performance became indistinguishable across laboratories (*Figure 3a–e*). We first examined the psychometric curves for the three sessions leading up to proficiency, which showed a stereotypical shape across mice and laboratories (*Figure 3a*). The average across mice of these psychometric curves was similar across laboratories (*Figure 3b*). The lapse rates (i.e., the errors made in response to easy contrasts of 50% and 100%) were low (9.5 ± 3.6%, *Figure 3c*) with no significant difference across laboratories. The slope of the curves, which measures contrast sensitivity, was also similar across laboratories, at 14.3 ± 3.8 (s.d., n = 7 laboratories, *Figure 3d*). Finally, the horizontal displacement of the curve, which measures response bias, was small at 0.3 ± 8.4 (s.d., n = 7, *Figure 3e*). None of these measures showed a significant difference across laboratories, either in median (performance: p=0.63, threshold: p=0.81, bias: p=0.81, FDR corrected Kruskal-Wallis test) or in variance (performance: p=0.09, threshold: p=0.57, bias: p=0.57, FDR corrected Levene's test). Indeed, mouse choices were no more consistent within labs than across labs (*Figure 3—figure supplement 1a*).

Variations across laboratories were also small in terms of trial duration and number of trials per session, even though we had made no specific effort to harmonize these variables. The median time from stimulus onset to feedback (trial duration, a coarse measure of reaction time) was 468 ± 221 ms, showing some differences across laboratories (*Figure 3—figure supplement 2a*, p=0.004, Kruskal-Wallis nonparametric test). Mice on average performed 719 ± 223 trials per session (*Figure 3—figure supplement 2b*). This difference was significant (p<$10^{-6}$, one-way ANOVA) but only in one laboratory relative to the rest.

Variation in performance across laboratories was so low that we were not able to assign a mouse to a laboratory based on performance (*Figure 3f*). Having found little variation in behavioral variables when considered one by one (*Figure 3c–e*), we asked whether mice from different laboratories may exhibit characteristic combinations of these variables. We thus trained a Naive Bayes classifier (*Pedregosa et al., 2011*), with 2000 random sub-samples of eight mice per laboratory, to predict lab membership from these behavioral variables. First, we established a positive control, checking that the classifier showed the expected behavior when provided with an informative variable: the time zone in which animals were trained. If the classifier was given this variable during training and testing it performed above chance (*Figure 3f*, *Positive control*) and the confusion matrix showed

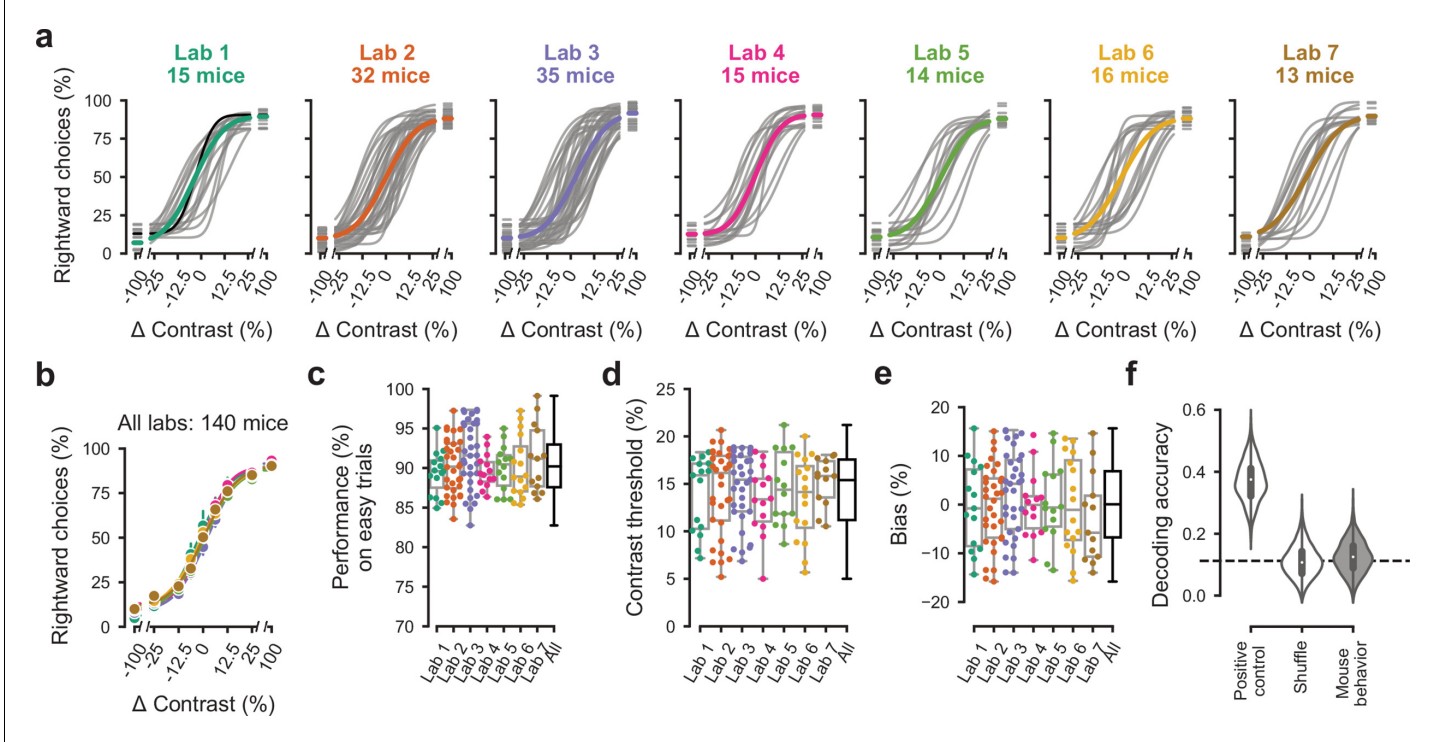

**Figure 3.** Performance in the basic task was indistinguishable across laboratories. (**a**) Psychometric curves across mice and laboratories for the three sessions at which mice achieved proficiency on the basic task. Each curve represents a mouse (*gray*). Black curve represents the example mouse in *Figure 1*. Thick colored lines show the lab average. (**b**) Average psychometric curve for each laboratory. Circles show the mean and error bars ± 68% CI. (**c**) performance on easy trials (50% and 100% contrasts) for each mouse, plotted per lab and over all labs. Colored dots show individual mice and boxplots show the median and quartiles of the distribution. (**d-e**) Same, for contrast threshold and bias. (**f**) Performance of a Naive Bayes classifier trained to predict from which lab mice belonged, based on the measures in (**c-e**). We included the timezone of the laboratory as a positive control and generated a null-distribution by shuffling the lab labels. Dashed line represents chance-level classification performance. *Violin plots*: distribution of the 2000 random sub-samples of eight mice per laboratory. *White dots*: median. *Thick lines*: interquartile range.

The online version of this article includes the following figure supplement(s) for figure 3:

**Figure supplement 1.** Mouse choices were no more consistent within labs than across labs.

**Figure supplement 2.** Behavioral metrics that were not explicitly harmonized showed small variation across labs.

**Figure supplement 3.** Classifiers could not predict lab membership from behavior.

**Figure supplement 4.** Comparable performance across institutions when using a reduced inclusion criterion (>=80% performance on easy trials).

**Figure supplement 5.** Behavior was indistinguishable across labs in the first 3 sessions of the full task.

clusters of labs located in the same time zone (*Figure 3—figure supplement 3b*). Next, we trained and tested the classifier on a null distribution obtained by shuffling the lab labels, and as expected we found that the classifier was at chance (*Figure 3f*, *Shuffle*). Finally, we trained and tested the classifier on the behavioral data, and found that it performed at chance level, failing to identify the laboratory of origin of the mice (*Figure 3f*, *Mouse behavior*). Its mean accuracy was 0.12 ± 0.051, and a 95th percentile of its distribution included the chance level of 0.11 (mean of the null distribution). The classifications were typically off-diagonal in the confusion matrix, and hence incorrect (*Figure 3—figure supplement 3c*). Similar results were obtained with two other classifier algorithms (*Figure 3—figure supplement 3d–i*).

This consistency across laboratories was not confined to the three sessions that led to proficiency, and was observed both in earlier and in later sessions. We repeated our analyses for three sessions that led to mice achieving a looser definition of proficiency: a single criterion of 80% correct for easy stimuli, without criteria on response bias and contrast threshold. In these three sessions, which could be earlier but not later than the ones analyzed in *Figure 3*, mouse behavior was again consistent across laboratories: a decoder failed to identify the origin of a mouse based on its behavioral performance (*Figure 3—figure supplement 4*). Finally, we repeated our analysis for sessions obtained

after the mouse achieved proficiency in the basic task. We selected the first three sessions of the full task, and again found no significant difference across laboratories: mice had similar performance at high contrast, similar contrast threshold, and similar response bias across laboratories (*Figure 3—figure supplement 5*).

## Performance in the full task was indistinguishable across laboratories

After training the mice in the purely sensory, basic task, we introduced them to the *full task*, where optimal performance requires integration of sensory perception with recent experience (*Figure 4a, b*). Specifically, we introduced block-wise biases in the probability of stimulus location, and therefore the more likely correct choice. Sessions started with a block of unbiased trials (50:50 probability of left vs. right) and then alternated between blocks of variable length (20–100 trials) biased toward the right (20:80 probability) or toward the left (80:20 probability) (*Figure 4a*). In these blocks, the probability of 0% contrast stimuli was doubled to match the probability of other contrasts. The transition between blocks was not signaled, so the mice had to estimate a prior for stimulus location based on recent task statistics. This task invites the mice to integrate information across trials and to use this prior knowledge in their perceptual decisions.

To assess how mice used information about block structure, we compared their psychometric curves in the different block types (*Figure 4b,c*). Mice incorporated block priors into their choices already from the first sessions in which they were exposed to this full task (*Figure 3—figure supplement 5a*). To assess proficiency in the full task, we used a fixed set of criteria (*Figure 1—figure supplement 1d*). We considered performance in the three sessions in which mice reached proficiency on the full task (*Figure 1—figure supplement 1d*). For mice that reached full task proficiency, the average training from start to finish took 31.5 ± 16.1 days, or 20,494 ± 10,980 trials (s.d., n = 98). The example mouse from Lab 1 (*Figure 2a*, black) took 19 days. The fastest learner achieved proficiency in 8 days (4691 trials), the slowest 81 days (61,316 trials). The curves for the 20:80 and 80:20 blocks were shifted relative to the curve for the 50:50 block, with mice more likely to choose right in the 20:80 blocks (where right stimuli appeared 80% of the time) and to choose left in the 80:20 blocks (where left stimuli appeared 80% of the time). As expected, block structure had the greatest impact on choices when sensory evidence was absent (contrast = 0%, *Figure 4c*). In these conditions, it makes sense for the mice to be guided by recent experience, and thus to choose differently depending on the block prior.

Changes in block type had a similar effect on mice in all laboratories (*Figure 4d,e*). The average shift in rightward choices was invariably highest at 0% contrast, where rightward choices in the two blocks differed by an average of 28.5%. This value did not significantly differ across laboratories any more than it differed within laboratories (one-way ANOVA $F_{(6)}$ = 1.345, p=0.2455, *Figure 4d,e*).

An analysis of the psychometric curves showed highly consistent effects of block type, with no significant differences across laboratories (*Figure 4f–i*). Changes in block type did not significantly affect the contrast threshold (Wilcoxon Signed-Rank test, p=0.85, n = 98 mice, *Figure 4f*). However, it did change the lapse rates, which were consistently higher on the left in the 20:80 blocks and on the right in the 80:20 blocks (Wilcoxon Signed-Rank test, lapse left; $p<10^{-6}$, lapse right; $p<10^{-7}$, *Figure 4g,h*). Finally, as expected there was a highly consistent change in overall bias, with curves shifting to the left in 20:80 trials and to the right in 80:20 trials (Wilcoxon Signed-Rank test, $p<10^{-16}$, *Figure 4i*). Just as in the basic task (*Figure 3*), a classifier trained on these variables could not predict above chance the origin laboratory of individual mice above chance (*Figure 4j*). Moreover, neither choice fractions nor bias shifts showed within-lab consistency that was larger than expected by chance (*Figure 3—figure supplement 1b,c*). This confirms that mice performed the full task similarly across laboratories.

## A probabilistic model reveals common strategies across mice and laboratories

Lastly, we investigated the strategies used by the mice in the basic task and in the full task and asked whether these strategies were similar across mice and laboratories. Mice might incorporate non-sensory information into their decisions, even when such information is not predictive of reward (*Busse et al., 2011*; *Lak et al., 2020a*). Therefore, even when reaching comparable performance levels, different mice might be weighing task variables differently when making a decision.

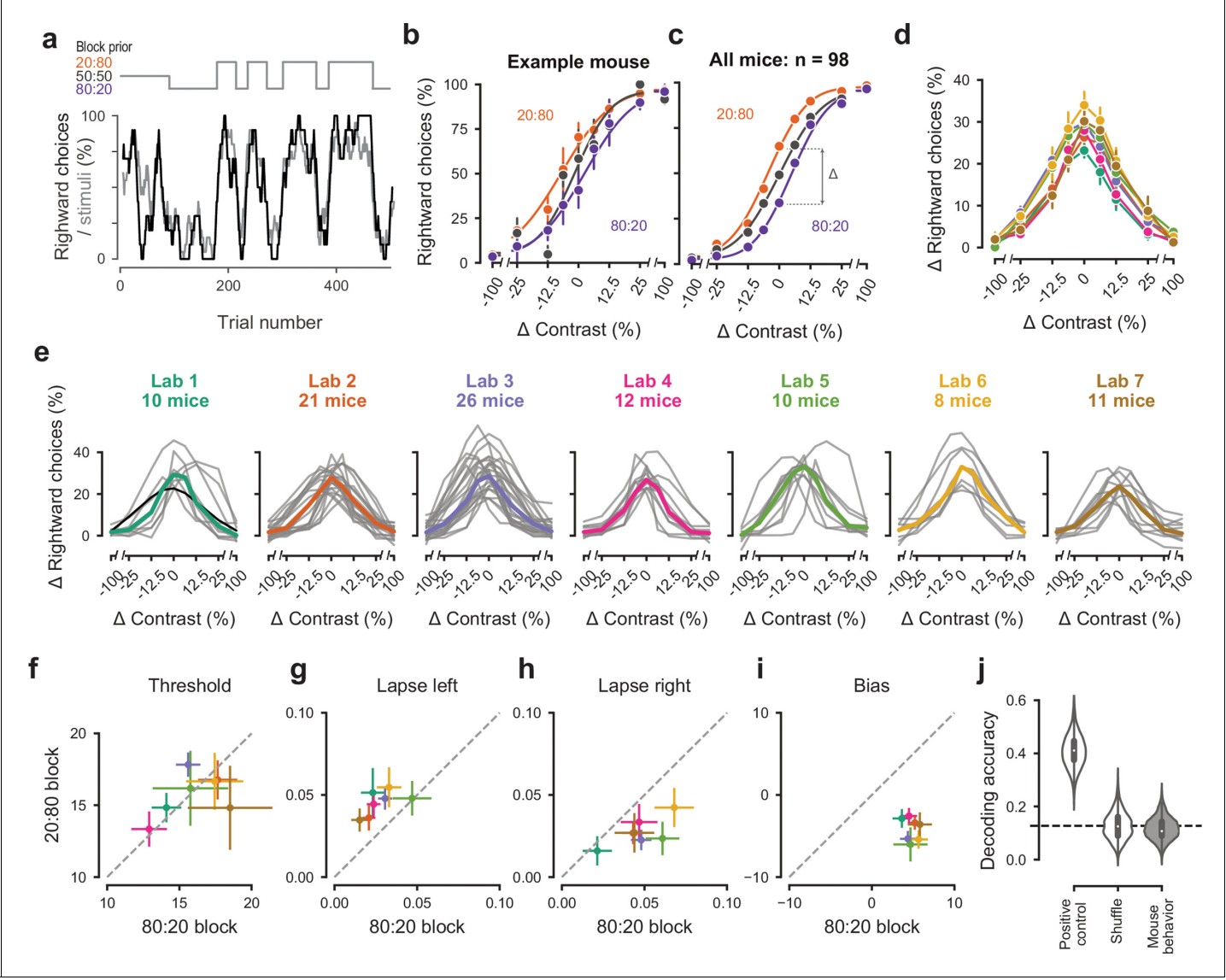

**Figure 4.** Mice successfully integrate priors into their decisions and task strategy. (a) Block structure in an example session. Each session started with 90 trials of 50:50 prior probability, followed by alternating 20:80 and 80:20 blocks of varying length. Presented stimuli (gray, 10-trial running average) and the mouse's choices (black, 10-trial running average) track the block structure. (b) Psychometric curves shift between blocks for the example mouse. (c) For each mouse that achieved proficiency on the full task (*Figure 1—figure supplement 1d*) and for each stimulus, we computed a 'bias shift' by reading out the difference in choice fraction between the 20:80 and 80:20 blocks (dashed lines). (d) Average shift in rightward choices between block types, as a function of contrast for each laboratory (colors as in 2 c, 3 c; error bars show mean ±68% CI). (e) Shift in rightward choices as a function of contrast, separately for each lab. Each line represents an individual mouse (gray), with the example mouse in black. Thick colored lines show the lab average. (f) Contrast threshold, (g) left lapses, (h) right lapses, and (i) bias separately for the 20:80 and 80:20 block types. Each lab is shown as mean +- s.e.m. (j) Classifier results as in 3 f, based on all data points in (f-i).

To quantify how different mice form their decisions, we used a generalized linear model (*Figure 5a*). The model is based on similar approaches used in both value-based (*Lau and Glimcher, 2005*) and sensory-based decision making (*Busse et al., 2011*; *Pinto et al., 2018*). In the model, the probability of making a right choice is calculated from a logistic function of the linear weighted sum of several predictors: the stimulus contrast, the outcome of the previous trial, and a bias term that represents the overall preference of a mouse for a particular choice across sessions (*Busse et al., 2011*). In the case of the full task, we added a parameter representing the identity of the block, measuring the weight of the prior stimulus statistics in the mouse decisions (*Figure 5a*).

We fitted the model to the choices of each individual mouse over three sessions by logistic regression. The mean condition number for the basic model was 2.4 and 3.2 for the full model. The low condition numbers for both models do not suffer from multicollinearity and the coefficients are therefore interpretable.

The model fit the mouse choices well and captured the relative importance of sensory and non-sensory information across mice and laboratories (*Figure 5b–g*). The model was able to accurately predict the behavior of individual mice, both in the basic task (*Figure 5b*) and in the full task (*Figure 5e*). As expected, visual terms had large weights, which grew with contrast to reach values above 3 at 100% contrast. Weights for non-sensory factors were much lower (*Figure 5c,f*, note different scale). Weights for past choices were positive for both rewarded (basic 0.19, full 0.42) and unrewarded previous trials (basic 0.33, full 0.46), suggesting that mice were perseverant in their choice behavior (*Figure 5d,g*; *Figure 5—figure supplement 1a*). Indeed, previous choices more strongly influenced behavior in the full task, both after rewarded ($t(125) = 10.736$, $p<10^{-6}$) and unrewarded trials ($t(125) = 4.817$, $p<10^{-6}$). Importantly, fitted weights and the model's predictive accuracy (unbiased: 81.03 + 5.1%, biased 82.1 + 5.8%) (*Figure 5—figure supplement 2*) were similar across laboratories, suggesting an overall common strategy.

The model coefficients demonstrated that mice were perseverant in their actions (*Figure 5d,g*; *Figure 5—figure supplement 1a*). This behavior can arise from insensitivity to the outcome of

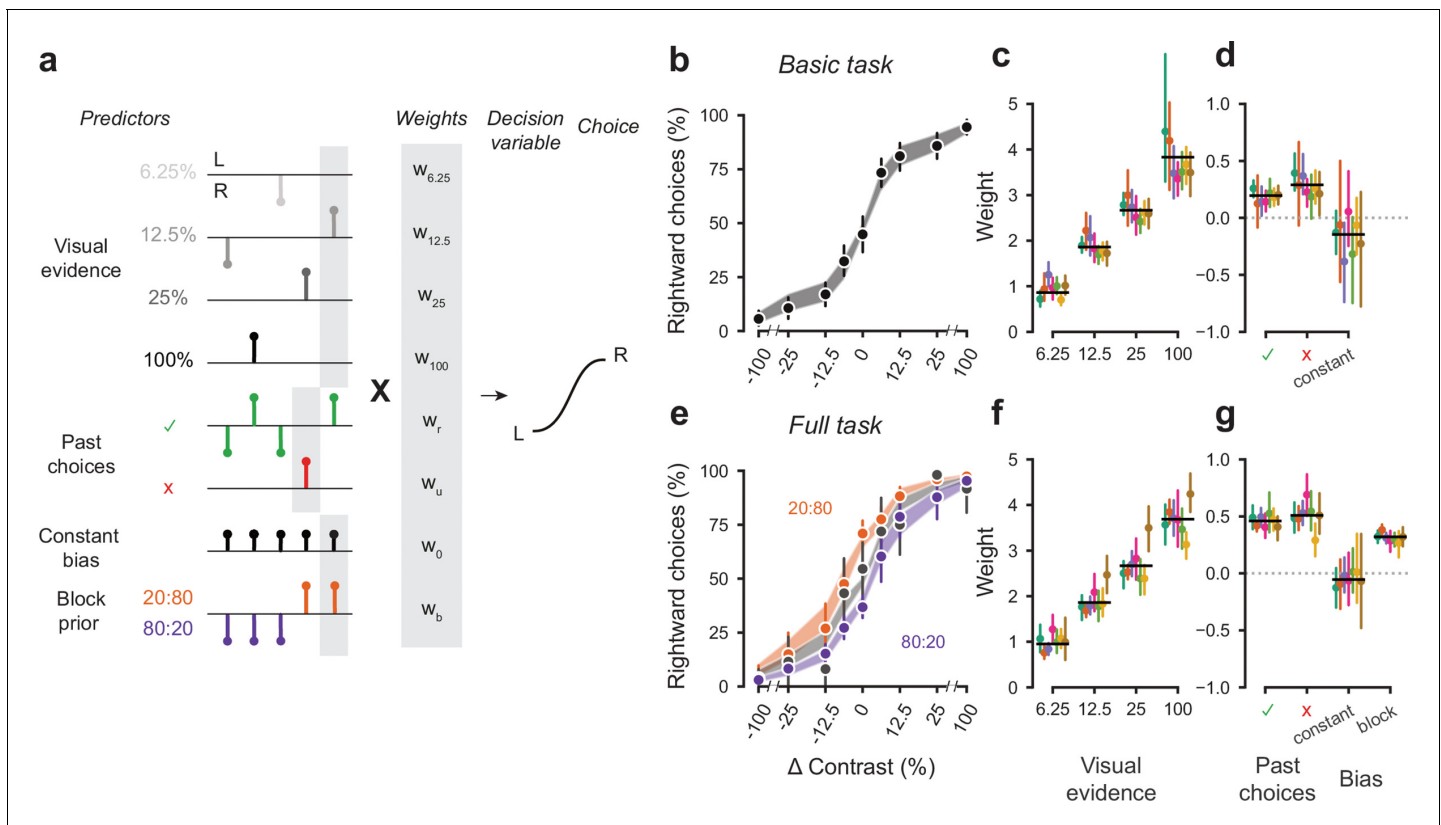

**Figure 5.** A probabilistic model reveals a common strategy across mice and laboratories. (**a**) Schematic diagram of predictors included in the GLM. Each stimulus contrast (except for 0%) was included as a separate predictor. Past choices were included separately for rewarded and unrewarded trials. The block prior predictor was used only to model data obtained in the full task. (**b**) Psychometric curves from the example mouse across three sessions in the basic task. Shadow represents 95% confidence interval of the predicted choice fraction of the model. Points and error bars represent the mean and across-session confidence interval of the data. (**c-d**) Weights for GLM predictors across labs in the basic task, error bars represent the 95% confidence interval across mice. (**e-g**), as b-d but for the full task.

The online version of this article includes the following figure supplement(s) for figure 5:

**Figure supplement 1.** History-dependent choice updating.

**Figure supplement 2.** Parameters of the GLM model of choice across labs.

previous trials and from slow drifts in the decision process across trials, arising from correlations in past choices independently of reward (*Lak et al., 2020a*; *Mendonça et al., 2018*). To disentangle these two factors, we corrected for slow across-trial drifts in the decision process (*Lak et al., 2020a*). This correction revealed a win-stay/lose-switch strategy in both the basic and full task, which coexists with slow drifts in choice bias across trials (*Figure 5—figure supplement 1b*). Moreover, the dependence on history- was modulated by confidence in the previous trial *Figure 5—figure supplement 1c* (*Lak et al., 2020a*; *Mendonça et al., 2018*). These effects were generally consistent across laboratories.

## Discussion

These results reveal that a complex mouse behavior can be successfully reproduced across laboratories, and more generally suggest a path toward improving reproducibility in neuroscience. To study mouse behavior across laboratories, we developed and implemented identical experimental equipment and a standard set of protocols. Not only did mice learn the task in all laboratories, but critically, after learning they performed the task comparably across laboratories. Mice in different laboratories had similar psychophysical performance in a purely sensory version of the task and adopted similar choice strategies in the full task, where they benefited from tracking the stimulus prior probability. Behavior showed variations across sessions and across mice, but these variations were no larger across laboratories than within laboratories.

Success did not seem guaranteed at the outset, because neuroscience faces a crisis of reproducibility (*Baker, 2016*; *Botvinik-Nezer et al., 2020*; *Button et al., 2013*) particularly when it comes to measurements of mouse behavior (*Chesler et al., 2002*; *Crabbe et al., 1999*; *Kafkafi et al., 2018*; *Sorge et al., 2014*; *Tuttle et al., 2018*). To solve this crisis, three solutions have been proposed: large studies, many teams, and upfront registration (*Ioannidis, 2005*). Our approach incorporates all three of these solutions. First, we collected vast amounts of data: 5 million choices from 140 mice. Second, we involved many teams, obtaining data in seven laboratories in three countries. Third, we standardized the experimental protocols and data analyses upfront, which is a key component of pre-registration.

An element that may have contributed to success is the collaborative, open-science nature of our initiative (*Wool and International Brain Laboratory, 2020*). Open-science collaborative approaches are increasingly taking hold in neuroscience (*Beraldo et al., 2019*; *Charles et al., 2020*; *Forscher et al., 2020*; *Koscielny et al., 2014*; *Poldrack and Gorgolewski, 2014*; *de Vries et al., 2020*). Our work benefited from collaborative development of the behavioral assay, and from frequent and regular meetings where data were reviewed across laboratories (*Figure 6*). These meetings helped identify problems at the origin, provide immediate feedback, and find collective solutions. Moreover, our work benefited from constant efforts at standardization. We took great care in standardizing and documenting the behavioral apparatus and the training protocol (see Appendices), to facilitate implementation across our laboratories and to encourage wider adoption by other laboratories. The protocols, hardware designs and software code are open-source and modular, allowing adjustments to accommodate a variety of scientific questions. The data are accessible at data.internationalbrainlab.org, and include all >5 million choices made by the mice.

Another element that might have contributed to success is our choice of behavioral task, which places substantial requirements on the mice while not being too complex. Previous failures to reproduce mouse behavior across laboratories typically arose in studies of unconstrained behavior such as responses to pain or stress (*Chesler et al., 2002*; *Crabbe et al., 1999*; *Kafkafi et al., 2018*; *Sorge et al., 2014*; *Tuttle et al., 2018*). Operant behaviors may be inherently more reproducible than the assays used in these studies. To be able to study decision making, and in hopes of achieving reproducibility, we designed a task that engages multiple brain processes from sensory perception and integration of evidence to combination of priors and evidence. It seems likely that reproducibility is easier to achieve if the task requirements are substantial (so there is less opportunity to engage in other behaviors) but not so complex that they fail to motivate and engage. Tasks that are too simple and unconstrained or too arbitrary and difficult may be hard to reproduce.

There are of course multiple ways to improve on our results, for example by clarifying, and if desired resolving, the differences in learning rate across mice, both within and across laboratories. The learning rate is a factor that we had not attempted to control, and we cannot here ascertain the

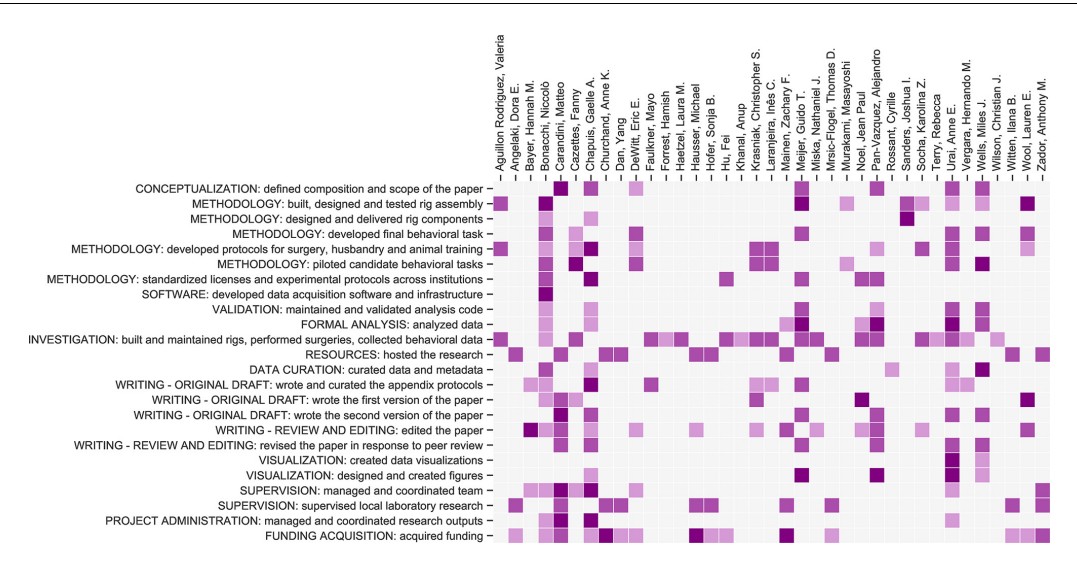

**Figure 6.** Contribution diagram. The following diagram illustrates the contributions of each author, based on the CRediT taxonomy (*Brand et al., 2015*). For each type of contribution there are three levels indicated by color in the diagram: 'support' (light), 'equal' (medium) and 'lead' (dark).

causes of its variability. We suspect that it might arise partly from variations in the expertise and familiarity of different labs with visual neuroscience and mouse behavior, which may impede standardization. If so, perhaps as experimenters gain further experience the differences in learning times will decrease. Indeed, an approach to standardizing learning rates might be to introduce full automation in behavioral training by reducing or even removing the need for human intervention (*Aoki et al., 2017*; *Poddar et al., 2013*; *Scott et al., 2015*). Variability in learning rates may be further reduced by individualized, dynamic training methods (*Bak et al., 2016*). If desired, such methods could also be aimed at obtaining uniform numbers of trials and reaction times across laboratories.

It would be fruitful to characterize the behavior beyond the turning of the wheel, analyzing the movement of the limbs. Here we only analyzed whether the wheel was turned to the left or to the right, but to turn the wheel the mice move substantial parts of their body, and they do so in diverse ways (e.g. some use two hands, others use one, and so on). Through videography (*Mathis et al., 2018*) one could classify these movements and perhaps identify behavioral strategies that provide insight in the performance of the task and into the diversity of behavior that we observed across sessions and across mice.

We hope that this large, curated dataset will serve as a benchmark for testing better models of decision-making. The model of choice that we used here, based on stimuli, choice history, reward history, and bias, is only a starting point (*Busse et al., 2011*). Indeed, the 'perseverance' that we represented by the reward history weights can arise from slow fluctuations in decision process over trials (*Lak et al., 2020a*; *Mendonça et al., 2018*) and history-dependence is modulated by confidence in the previous trial (*Lak et al., 2020b*; *Lak et al., 2020a*; *Urai et al., 2017*). In addition, our data can help investigate phenomena such as the origin of lapses (*Ashwood et al., 2020*; *Pisupati et al., 2021*), the tracking of changes between prior blocks (*Norton et al., 2019*), the effect of fluctuating engagement states (*McGinley et al., 2015*), and the dynamics of trial-by-trial learning (*Roy et al., 2021*).

To encourage and support community adoption of this task, we provide detailed methods and protocols. These methods and protocols provide a path to training mice in this task and reproducing their behavior across laboratories, but of course we cannot claim that they constitute the optimal path. In designing our methods, we made many choices that were based on intuition rather than rigorous experimentation. We don't know what is crucial in these methods and what is not.

The reproducibility of this mouse behavior makes it a good candidate for studies of the brain mechanisms underlying decision making. A reproducible behavioral task can be invaluable to

establish the neural basis of behavior. If different studies use the same task, they can directly compare their findings. There are indeed illustrious examples of behavioral tasks that serve this role. For studying decision-making in primates, these include the tactile flutter comparison task (*de Lafuente and Romo, 2005*; *Romo et al., 2012*) and the random dots visual discrimination task (*Ding and Gold, 2013*; *Newsome et al., 1989*; *Shadlen and Kiani, 2013*). Both tasks have been used in multiple studies to record from different brain regions while enabling a meaningful comparison of the results. Conversely, without a standardized behavioral task we face the common situation where different laboratories record from different neurons in different regions in different tasks, likely drawing different conclusions and likely not sharing their data. In that situation it is not possible to establish which factors determine the different conclusions and come to a collective understanding.

Now that we have developed this task and established its reproducibility across laboratories, the International Brain Laboratory is using it together with neural recordings, which are performed in different laboratories and combined into a single large data set. Other laboratories that adopt this task for studies of neural function will then be able to rely on this large neural dataset to complement their more focused results. Moreover, they will be able to compare each other's results, knowing that any difference between them is unlikely to be due to differences in behavior. We also hope that these resources catalyze the development of new adaptations and variations of our approach, and accelerate the use of mice in high quality, reproducible studies of neural correlates of decision-making.

## Materials and methods

All procedures and experiments were carried out in accordance with the local laws and following approval by the relevant institutions: the Animal Welfare Ethical Review Body of University College London; the Institutional Animal Care and Use Committees of Cold Spring Harbor Laboratory, Princeton University, and University of California at Berkeley; the University Animal Welfare Committee of New York University; and the Portuguese Veterinary General Board.

### Animals

Animals (all female and male C57BL6/J mice aged 3–7 months obtained from Jackson Laboratory or Charles River) were co-housed whenever possible, with a minimum enrichment of nesting material and a mouse house. Mice were kept in a 12 hr light-dark cycle, and fed with food that was 5–6% fat and 18–20% protein. See *Appendix 1—table 1* for details on standardization.

### Surgery

A detailed account of the surgical methods is in Protocol 1 (*The International Brain Laboratory, 2020a*). Briefly, mice were anesthetized with isoflurane and head-fixed in a stereotaxic frame. The hair was then removed from their scalp, much of the scalp and underlying periosteum was removed and bregma and lambda were marked. Then the head was positioned such that there was a 0 degree angle between bregma and lambda in all directions. The headbar was then placed in one of three stereotactically defined locations and cemented in place. The exposed skull was then covered with cement and clear UV curing glue, ensuring that the remaining scalp was unable to retract from the implant.

### Materials and apparatus

For detailed parts lists and installation instructions, see Protocol 3 (*The International Brain Laboratory, 2021a*). Briefly, all labs installed standardized behavioral rigs consisting of an LCD screen (LP097Q $\times$ 1, LG), a custom 3D-printed mouse holder and head bar fixation clamp to hold a mouse such that its forepaws rest on a steering wheel (86652 and 32019, LEGO) (*Burgess et al., 2017*). Silicone tubing controlled by a pinch valve (225P011-21, NResearch) was used to deliver water rewards to the mouse. The general structure of the rig was constructed from Thorlabs parts and was placed inside an acoustical cabinet (9U acoustic wall cabinet 600 $\times$ 600, Orion). To measure the precise times of changes in the visual stimulus (which is important for future neural recordings), a patch of pixels on the LCD screen flipped between white and black at every stimulus change, and this flip was captured with a photodiode (Bpod Frame2TTL, Sanworks). Ambient temperature, humidity, and barometric air pressure were measured with the Bpod Ambient module (Sanworks), wheel position

was monitored with a rotary encoder (05.2400.1122.1024, Kubler) connected to a Bpod Rotary Encoder Module (Sanworks). Video of the mouse was recorded with a USB camera (CM3-U3-13Y3M-CS, Point Grey). A speaker (HPD-40N16PET00-32, Peerless by Tymphany) was used to play task-related sounds, and an ultrasonic microphone (Ultramic UM200K, Dodotronic) was used to record ambient noise from the rig. All task-related data was coordinated by a Bpod State Machine (Sanworks). The task logic was programmed in Python and the visual stimulus presentation and video capture was handled by Bonsai (*Lopes et al., 2015*) and the Bonsai package BonVision (*Lopes et al., 2021*).

## Habituation, training, and experimental protocol

For a detailed protocol on animal training, see Protocol 2 (*The International Brain Laboratory, 2020b*). Mice were water restricted or given access to citric acid water on weekends. The advantage of the latter solution is that it does not require measuring precise amounts of fluids during the weekend (*Urai et al., 2021*). Mice were handled for at least 10 min and given water in hand for at least two consecutive days prior to head fixation. On the second of these days, mice were also allowed to freely explore the rig for 10 min. Subsequently, mice were gradually habituated to head fixation over three consecutive days (15–20, 20–40, and 60 min, respectively), observing an association between the visual grating and the reward location. On each trial, with the steering wheel locked, mice passively viewed a Gabor stimulus (100% contrast, 0.1 cycles/degree spatial frequency, random phase, vertical orientation) presented on a small screen (size: approx. 246 mm diagonal active display area). The screen was positioned 8 cm in front of the animal and centralized relative to the position of eyes to cover ~102 visual degree azimuth. The stimulus appeared for ~10 s randomly presented at −35° (left),+35° (right), or 0° (center) and the mouse received a reward in the latter case (3 μl water with 10% sucrose).

On the fourth day, the steering wheel was unlocked and coupled to the movement of the stimulus. For each trial, the mouse must use the wheel to move the stimulus from its initial location to the center to receive a reward. Initially, the stimulus moved 8°/mm of movement at the wheel surface. Once the mouse completed at least 200 trials within a session, the gain of the wheel for the following sessions was halved, to at 4°/mm. At the beginning of each trial, the mouse was required to not move the wheel for a quiescence period of 400–700 ms (randomly drawn from an exponential distribution with a mean of 550 ms). If the wheel moved during this period, the timer was reset. After the quiescence period, the stimulus appeared on either the left or right (±35° azimuth, within the field of binocular vision in mice, *Seabrook et al., 2017*) with a contrast randomly selected from a predefined set (initially, 50% and 100%). Simultaneously, an onset tone (5 kHz sine wave, 10 ms ramp) was played for 100 ms. When the stimulus appeared, the mouse had 60 s to move it. A response was registered if the center of the stimulus crossed the ±35 deg azimuth line from its original position (simple threshold crossing, no holding period required) (*Burgess et al., 2017*). If the mouse correctly moved the stimulus 35° to the center of the screen, it immediately received a 3 μL reward; if it incorrectly moved the stimulus 35° away from the center (20° visible and the rest off-screen), it received a timeout. Reward delivery was therefore not contingent on licking, and as such licking was not monitored online during the task. If the mouse responded incorrectly or failed to reach either threshold within the 60 s window, a noise burst was played for 500 ms and the inter-trial interval was set to 2 s. If the response was incorrect and the contrast was 'easy' ($\geq$50%), a 'repeat' trial followed, in which the previous stimulus contrast and location was presented with a high probability (see Protocol 2 [*The International Brain Laboratory, 2020b*]). When in the rig, the animal was monitored via a camera to ensure the experiment was proceeding well (e.g. lick spout reachable, animal engaged).

To declare a mouse proficient in the basic task ('Level 1') we used two consecutive sets of criteria. The first set, called '1a', were introduced in September 2018, before training started on the mice that appear in this paper (January 2019). These criteria were obtained by analyzing the data of *Lak et al., 2020b* and were verified on pilot data acquired in 3 of our labs. The second set, called '1b', was introduced shortly afterwards (September 2019) and was applied to 60 of the 140 mice. It was more stringent, to offset possible decreases in performance that may occur during subsequent neural recordings. For a thorough definition of the training criteria, see Appendix 2 (*The International Brain Laboratory, 2020b*) section 'Criteria to assess learning >Trained'. Briefly, the 1a/1b criteria called for the mouse to reach the following targets in three consecutive sessions:

200/400 completed trials; performance on easy trials > 80%/90%; and fitted psychometric curves with absolute bias <16%/10%, contrast threshold <19%/20%, lapse rates < 0.2/0.1. Additionally, the 1b criteria required for median reaction times across the three sessions for the 0% contrast trials to be <2 s.

Out of the 206 mice that we trained, 66 did not complete Level one for the following reasons: 16 mice died or were culled due to infection, illness or injury; 17 mice could not be trained due to experimental impediments (e.g. too many mice in the pipeline, experimenter ill, broken equipment); 14 mice did not learn in 40 days of training due to extremely high bias and/or low trial count (n = 9) or an otherwise low performance (n = 5); 12 mice reached at least the first level within 40 days but progressed too slowly or were too old. For the remaining seven mice, the reason was undocumented.

Once an animal was proficient in the basic task, it proceeded to the full task. Here, the trial structure was identical, except that stimuli were more likely to reappear on the same side for variable blocks of trials, and counterbiasing 'repeat' trials were not used. Each session began with 90 trials in which stimuli were equally likely to appear on the left or right (10 repetitions at each contrast), after which the probability of the stimulus appearing on the left alternated between 0.8 and 0.2 for a given block. The number of trials in each block was on average 51, and was drawn from a geometric distribution, truncated to have values between 20 and 100.

To declare a mouse proficient in the full task, its performance was assessed using three successive sessions (*Figure 1—figure supplement 1d*). For each of the sessions, the mouse had to perform at least 400 trials, with a performance of at least 90% correct on 100% contrast trials. Also, using a combination of all the trials of the three sessions, the lapses (both left and right, and for each of the block types) had to be below 0.1, the bias above 5, and the median reaction time on 0% contrast over 2 s.

## Psychometric curves

To obtain a psychometric curve, the data points were fitted with the following parametric error function, using a maximum likelihood procedure:

$$P = \gamma + (1 - \gamma - \lambda)\, erf\left(\frac{c - \mu}{\sigma} + 1\right) / 2$$

Where $P$ is the probability of a rightward choice, $c$ is the stimulus contrast, and the rest are fitted parameters:

$\gamma$ is the lapse rate for left stimuli
$\lambda$ is the lapse rate for right stimuli
$\mu$ is the response bias
$\sigma$ is the contrast threshold.

The procedures to fit these curves and obtain these parameters are described in detail in Protocol 2 (*The International Brain Laboratory, 2020b*).

## Classification of laboratory membership

Three different classifiers were used to try to predict in which laboratory a mouse was trained based on behavioral metrics: Naive Bayes, Random Forest, and Logistic Regression. We used the scikit-learn implementation available in Python with default configuration settings for the three classifiers. Some labs trained more mice than others resulting in an imbalanced dataset. This imbalance was corrected by taking a random subsample of 8 mice from each laboratory 2000 times. The size of the subsample was chosen because eight mice was the lowest number of mice over all different datasets for which classification was performed. For each random subsample, lab membership was classified using leave-one-out cross-validation. Furthermore, a null-distribution was generated by shuffling the lab labels for each subsample and classifying the shuffled data. The classification accuracy was calculated as:

$$accuracy = \frac{number\ of\ correctly\ classified\ mice}{total\ number\ of\ mice} \tag{1}$$

Here the total number of mice was 56 because eight mice were randomly subsampled from seven labs.

## Probabilistic choice model

To quantify and describe the different factors affecting choices across labs, we adapted a probabilistic choice model *Busse et al., 2011* used in a similar task. The model is a binomial logistic regression model, where the observer estimates the probability of choosing right ($p$) or left ($1-p$) from sensory and non-sensory information. In the model, probabilities are obtained from the logistic transformation of the decision variable $z$ (*Equation 2*), which in itself is the result of a weighted linear function of different task predictors (*Equation 3* and *Equation 4*).

$$p = \frac{1}{1 + e^{-z}} \tag{2}$$

For the *basic task*, in each trial $i$, the decision variable $z$ is calculated by:

$$z(i) = \sum_c W_c I_c(i) + W_r r(i-1) + W_u u(i-1) + W_0 \tag{3}$$

where $W_c$ is the coefficient associated with the contrast $c \in \{6.25, 12.5, 25, 50, 100\}$, and $I_c(i)$ is an indicator function indicating +1 if the contrast $c$ appeared on the right in trial $i$, -1 if it appeared on the left and 0 if that contrast was not presented. The coefficients ($W_r$ and $W_u$) weigh the effect of previous choices, depending on their outcome. $r(i-1)$ is defined as +1 when the previous trial was on the right and rewarded, -1 when on the left and rewarded, and 0 when unrewarded. Conversely, $u(i-1)$ is defined as +1 when the previous trial was on the right and unrewarded, -1 when on the left and unrewarded and 0 when rewarded. $W_0$ is a constant representing the overall bias of the mouse.

When modelling the *full task* we also included the term $W_b b(i)$ (*Equation 4*), which captures the block identity $b(i)$ for trial $i$. $b(i)$ is defined as +1 if trial $i$ is part of an 20:80 block, -1 if part of a 80:20 block and 0 if part of a 50/50 block:

$$z(i) = \sum_c W_c I_c(i) + W_r r(i-1) + W_u u(i-1) + W_0 + W_b b(i) \tag{4}$$

For each animal, the design matrix was built using patsy (*Smith et al., 2018*). The model was then fitted by regularized maximum likelihood estimation, using the *Logit.fit_regularized* function in statsmodels (*Seabold and Perktold, 2010*). For the example animal (*Figure 5b,e*), 10,000 samples were drawn from a multivariate Gaussian (obtained from the inverse of the model's Hessian), and for each sample the model 's choice fraction for each contrast level was predicted. Confidence intervals were then defined as the 0.025 and 0.975 quantiles across samples.

## Data and code availability

All data presented in this paper is publicly available. It can be viewed and accessed in two ways: via DataJoint and web browser tools at data.internationalbrainlab.org.

All data were analyzed and visualized in Python, using numpy (*Harris et al., 2020*), pandas (*Reback et al., 2020*) and seaborn (*Waskom, 2021*). Code to produce all the figures is available at github.com/int-brain-lab/paper-behavior (copy archived at swh:1:rev: edc453189104a1f76f4b2ab230cd86f2140e3f63; *The International Brain Laboratory, 2021b*) and a Jupyter notebook for re-creating *Figure 2* can be found at https://jupyterhub.internationalbrainlab. com.

## Acknowledgements

We thank Charu Reddy for helping develop animal welfare and surgical procedures; George Bekheet, Filipe Carvalho, Paulo Carriço, Robb Barrett and Del Halpin for help with hardware design; Luigi Acerbi and Zoe Ashwood for advice about model fitting; and Peter Dayan and Karel Svoboda for comments on the manuscript. AEU is supported by the German National Academy of Sciences Leopoldina. LEW is supported by a Marie Skłodowska-Curie Actions fellowship. FC was supported

by an EMBO long term fellowship and an AXA postdoctoral fellowship. HMV was supported by an EMBO long term fellowship. MC holds the GlaxoSmithKline/Fight for Sight Chair in Visual Neuroscience. This work was supported by grants from the Wellcome Trust (209558 and 216324) and the Simons Foundation. The production of all IBL Platform Papers is led by a Task Force, which defines the scope and composition of the paper, assigns and/or performs the required work for the paper, and ensures that the paper is completed in a timely fashion. The Task Force members for this platform paper are Gaelle A Chapuis, Guido T Meijer, Alejandro Pan Vazquez, Anne E Urai, Miles Wells, and Matteo Carandini.

## Additional information

### Competing interests

Joshua Sanders: JIS is the owner of Sanworks LLC which provides hardware and consulting for the experimental set-up described in this work. The other authors declare that no competing interests exist.

### Funding

| Funder | Grant reference number | Author |
|---|---|---|
| Wellcome Trust | 209558 | Dora Angelaki<br>Matteo Carandini<br>Anne K Churchland<br>Yang Dan<br>Michael Häusser<br>Sonja B Hofer<br>Zachary F Mainen<br>Thomas D Mrsic-Flogel<br>Ilana B Witten<br>Anthony M Zador |
| Simons Foundation | | Dora Angelaki<br>Matteo Carandini<br>Anne K Churchland<br>Yang Dan<br>Michael Häusser<br>Sonja B Hofer<br>Zachary F Mainen<br>Thomas D Mrsic-Flogel<br>Ilana B Witten<br>Anthony M Zador |
| Wellcome Trust | 216324 | Dora Angelaki<br>Matteo Carandini<br>Anne K Churchland<br>Yang Dan<br>Michael Häusser<br>Sonja B Hofer<br>Zachary F Mainen<br>Thomas D Mrsic-Flogel<br>Ilana B Witten<br>Anthony M Zador |
| German National Academy of Sciences Leopoldina | | Anne E Urai |
| Marie Skłodowska-Curie Actions, European Commission | | Lauren E Wool |
| EMBO | Long term fellowship | Fanny Cazettes<br>Hernando Vergara |
| AXA Research Fund | Postdoctoral fellowship | Fanny Cazettes |

The funders had no role in study design, data collection and interpretation, or the decision to submit the work for publication.

## Author contributions

Valeria Aguillon-Rodriguez, Methodology: built, designed and tested rig assembly (equal); developed protocols for surgery, husbandry and animal training (equal); Investigation: built and maintained rigs, performed surgeries, collected behavioral data (equal); Dora Angelaki, Resources: hosted the research (equal); Supervision: supervised local laboratory research (equal); Funding Acquisition: acquired funding (support); Hannah Bayer, Writing - Original draft: wrote and curated the appendix protocols (support); Writing - Review and editing: edited the paper (lead); Supervision: managed and coordinated team (support); Niccolo Bonacchi, Methodology: built, designed and tested rig assembly (lead); designed and delivered rig components (support); piloted candidate behavioral tasks (equal); developed final behavioral task (equal); developed protocols for surgery, husbandry and animal training (support); standardized licenses and experimental protocols across institutions (equal); Software: developed data acquisition software and infrastructure (lead); Validation: maintained and validated analysis code (support); Formal analysis: analyzed data (support); Invetigation: built and maintained rigs, performed surgeries, collected behavioral data (support); Data curation: curated data and metadata (equal); Writing - Original draft: wrote the first version of the paper (support); wrote and curated the appendix protocols (support); Writing - Review and editing: edited the paper (support); Supervision: managed and coordinated team (support); Project administration: managed and coordinated research outputs (support); Funding acquisition: acquired funding (support); Matteo Carandini, Conceptualization: defined composition and scope of the paper (lead); Resources: hosted the research (equal); Writing - Original draft: wrote the first version of the paper (equal); wrote the second version of the paper (lead); Writing - Review and editing: edited the paper (equal); Supervision: supervised local laboratory research (equal); managed and coordinated team (lead); Project administration: managed and coordinated research outputs (lead); Funding acquisition: acquired funding (equal); Fanny Cazettes, Methodology: piloted candidate behavioral tasks (lead); developed final behavioral task (support); developed protocols for surgery, husbandry and animal training (support); Investigation: built and maintained rigs, performed surgeries, collected behavioral data (equal); Writing - Original draft: wrote the first version of the paper (support); Supervision: managed and coordinated team (support); Gaelle Chapuis, Conceptualization: defined composition and scope of the paper (equal); Methodology: developed protocols for surgery, husbandry and animal training (lead); designed and delivered rig components (support); standardized licenses and experimental protocols across institutions (lead); Validation: maintained and validated analysis code (support); Formal analysis: analyzed data (support); Data curation: curated data and metadata (support); Writing - Original draft: wrote the second version of the paper (equal); wrote and curated the appendix protocols (lead); Writing - Review and editing: edited the paper (support); Visualization: designed and created figures (support); Supervision: managed and coordinated team (lead); Project administration: managed and coordinated research outputs (lead); Funding acqusition: acquired funding (support); Anne K Churchland, Resources: hosted the research (equal); Supervision: supervised local laboratory research (equal); Funding acquistion: acquired funding (lead); Yang Dan, Sonja B Hofer, Thomas D Mrsic-Flogel, Ilana B Witten, Resources: hosted the research (equal); Supervision: supervised local laboratory research (equal); Funding acquisition: acquired funding (support); Eric Dewitt, Conceptualization: defined composition and scope of the paper (support); Methodology: developed final behavioral task (equal); piloted candidate behavioral tasks (equal); developed protocols for surgery, husbandry and animal training (support); Writing - Review and editing: edited the paper (support); Supervision: managed and coordinated team (support); Funding acquisition: acquired funding (support); Mayo Faulkner, Investigation: built and maintained rigs, performed surgeries, collected behavioral data (equal); Writing - Original draft: wrote and curated the appendix protocols (equal); Hamish Forrest, Anup Khanal, Rebecca Terry, Christian J Wilson, Investigation: built and maintained rigs, performed surgeries, collected behavioral data (support); Laura Haetzel, Investigation: built and maintained rigs, performed surgeries, collected behavioral data (equal); Michael Häusser, Resources: hosted the research (equal); Writing - Review and editing: edited the paper (support); Supervision: supervised local laboratory research (equal); Funding acquisition: acquired funding (lead); Fei Hu, Methodology: standardized licenses and experimental protocols across institutions (equal); Investigation: built and maintained rigs, performed surgeries, collected behavioral data (equal); Funding acquisition: acquired funding (support); Christopher Krasniak, Methodology: piloted candidate behavioral tasks (equal); developed protocols for surgery, husbandry and animal training (equal); Investigation: built and maintained rigs,

performed surgeries, collected behavioral data (equal); Writing - Original draft: wrote the first version of the paper (equal); wrote and curated the appendix protocols (support); Writing - Review and editing: edited the paper (support); Ines Laranjeira, Methodology: piloted candidate behavioral tasks (equal); developed protocols for surgery, husbandry and animal training (equal); Investigation: built and maintained rigs, performed surgeries, collected behavioral data (equal); Writing - Original draft: wrote and curated the appendix protocols (support); Zachary F Mainen, Formal analysis: analyzed data (support); Resources: hosted the research (equal); Writing - Review and editing: edited the paper (equal); Supervision: supervised local laboratory research (equal); Funding acquisition: acquired funding (lead); Guido Meijer, Conceptualization: defined composition and scope of the paper (equal); Methodology: developed final behavioral task (equal); built, designed and tested rig assembly (lead); standardized licenses and experimental protocols across institutions (equal); Validation: maintained and validated analysis code (equal); Formal analysis: analyzed data (lead); Investigation: built and maintained rigs, performed surgeries, collected behavioral data (equal); Writing - Original draft: wrote the second version of the paper (equal); wrote and curated the appendix protocols (equal); Visualization: designed and created figures (lead); Nathaniel J Miska, Investigation: built and maintained rigs, performed surgeries, collected behavioral data (equal); Writing - Review and editing: edited the paper (support); Masayoshi Murakami, Methodology: built, designed and tested rig assembly (support); piloted candidate behavioral tasks (support); Jean-Paul Noel, Methodology: standardized licenses and experimental protocols across institutions (equal); Formal analysis: analyzed data (support); Investigation: built and maintained rigs, performed surgeries, collected behavioral data (equal); Writing - Original draft: wrote the first version of the paper (lead); Writing - Review and editing: edited the paper (support); Alejandro Pan-Vazquez, Conceptualization: defined composition and scope of the paper (equal); Methodology: standardized licenses and experimental protocols across institutions (equal); developed protocols for surgery, husbandry and animal training (support); Validation: maintained and validated analysis code (support); Formal analysis: analyzed data (lead); Investigation: built and maintained rigs, performed surgeries, collected behavioral data (equal); Writing - Original draft: wrote the second version of the paper (equal); Writing - Review and editing: edited the paper (equal); Visualization: designed and created figures (lead); Cyrille Rossant, Data curation: curated data and metadata (support); Joshua Sanders, Methodology: designed and delivered rig components (lead); built, designed and tested rig assembly (equal); Karolina Socha, Methodology: developed protocols for surgery, husbandry and animal training (equal); built, designed and tested rig assembly (support); Investigation: built and maintained rigs, performed surgeries, collected behavioral data (equal); Writing - Review and editing: edited the paper (support); Anne E Urai, Conceptualization: defined composition and scope of the paper (equal); Methodology: built, designed and tested rig assembly (support); piloted candidate behavioral tasks (equal); developed final behavioral task (equal); developed protocols for surgery, husbandry and animal training (equal); Validation: maintained and validated analysis code (equal); Formal analysis: analyzed data (lead); Investigation: built and maintained rigs, performed surgeries, collected behavioral data (equal); Data curation: curated data and metadata (support); Writing - Original draft: wrote the second version of the paper (equal); wrote and curated the appendix protocols (support); Visualization: designed and created figures (lead); created data visualizations (lead); Supervision: managed and coordinated team (support); Project administration: managed and coordinated research outputs (support); Hernando Vergara, Investigation: built and maintained rigs, performed surgeries, collected behavioral data (support); Writing - Original draft: wrote and curated the appendix protocols (support); Miles Wells, Conceptualization: defined composition and scope of the paper (equal); Methodology: built, designed and tested rig assembly (support); piloted candidate behavioral tasks (lead); developed final behavioral task (equal); Validation: maintained and validated analysis code (equal); Formal analysis: analyzed data (equal); Data curation: curated data and metadata (lead); Writing - Original draft: wrote the second version of the paper (equal); Visualization: designed and created figures (support); created data visualizations (support); Lauren E Wool, Methodology: built, designed and tested rig assembly (lead); developed final behavioral task (equal); developed protocols for surgery, husbandry and animal training (support); Writing - Original draft: wrote the first version of the paper (lead); Writing - Review and editing: edited the paper (equal); Funding acquisition: acquired funding (support); Anthony M Zador, Resources: hosted the research (equal); Supervision: supervised local laboratory research (equal); managed and coordinated team (equal); Funding acquisition: acquired funding (equal)

## Author ORCIDs

Dora Angelaki (iD) https://orcid.org/0000-0002-9650-8962
Hannah Bayer (iD) https://orcid.org/0000-0002-5644-4124
Matteo Carandini (iD) https://orcid.org/0000-0003-4880-7682
Fanny Cazettes (iD) https://orcid.org/0000-0002-9648-4761
Anne K Churchland (iD) https://orcid.org/0000-0002-3205-3794
Yang Dan (iD) https://orcid.org/0000-0002-3818-877X
Michael Häusser (iD) https://orcid.org/0000-0002-2673-8957
Fei Hu (iD) https://orcid.org/0000-0001-7827-9548
Anup Khanal (iD) https://orcid.org/0000-0002-8929-7984
Zachary F Mainen (iD) https://orcid.org/0000-0001-7913-9109
Jean-Paul Noel (iD) https://orcid.org/0000-0001-5297-3363
Anne E Urai (iD) https://orcid.org/0000-0001-5270-6513
Ilana B Witten (iD) https://orcid.org/0000-0003-0548-2160

## Ethics

Animal experimentation: All procedures and experiments were carried out in accordance with the local laws and following approval by the relevant institutions: the Animal Welfare Ethical Review Body of University College London [P1DB285D8]; the Institutional Animal Care and Use Committees of Cold Spring Harbor Laboratory [1411117; 19.5], Princeton University [1876-20], and University of California at Berkeley [AUP-2016-06-8860-1]; the University Animal Welfare Committee of New York University [18-1502]; and the Portuguese Veterinary General Board [0421/0000/0000/2016-2019].

## Decision letter and Author response

Decision letter https://doi.org/10.7554/eLife.63711.sa1
Author response https://doi.org/10.7554/eLife.63711.sa2

# Additional files

## Supplementary files

- Transparent reporting form

## Data availability

Data for all figures is available at https://data.internationalbrainlab.org/.

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

## Appendix 1

## Standardization

**Appendix 1—table 1.** Standardization.
a, To facilitate reproducibility we standardized multiple aspects of the experiment. Some variables were kept strictly the same across mice, while others were kept within a range or simply recorded (see 'Standardized' column). b-c, The behavior training protocol was also standardized. Several task parameters adaptively changed within or across sessions contingent on various performance criteria being met, including number of trials completed, amount of water received and proportion of correct responses.

### A

| Category | Variable | Standardized | Standard | Recorded |
|---|---|---|---|---|
| Animal | Weight | Within a range | 18–30 g at headbar implant | Per session |
| | Age | Within a range | 10–12 weeks at headbar implant | Per session |
| | Strain | Exactly | C57BL/6J | Once |
| | Sex | No | Both | Once |
| | Provider | Two options | Charles River (EU) Jax (US) | Once |
| Training | Handling | One protocol | Protocol 2 | No |
| | Hardware | Exactly | Protocol 3 | No |
| | Software | Exactly | Protocol 3 | Per session |
| | Fecal count | N/A | N/A | Per session |
| | Time of day | No | As constant as possible | Per session |
| Housing | Enrichment | Minimum requirement | At least nesting and house | Once |
| | Food | Within a range | Protein: 18–20%, Fat: 5–6.2% | Once |
| | Light cycle | Two options | 12 Hr inverted or non-inverted | Once |
| | Weekend water | Two options | Citric acid water or measured water | Per session |
| | Co housing status | No | Co-housing preferred, separate problem mice | Per change |
| Surgery | Aseptic protocols | One protocol | Protocol 1 | No |
| | Tools/Consumables | Required parts | Protocol 1 | No |

### B

*Continued on next page*

| Adaptive parameter | Initial value |
|---|---|
| Contrast set | [100, 50] |
| Reward volume | 3 µL |
| Wheel gain | 8 deg/mm |

**C**

| Criterion | Outcome |
|---|---|
| >200 trials completed in previous session | Wheel gain decreased 4 deg/mm |
| >80% correct on each contrast | Contrast set = [100, 50, 25] |
| >80% correct on each contrast after above | Contrast set = [100, 50, 25, 12.5] |
| 200 trials after above | Contrast set = [100, 50, 25, 12.5, 6.25] |
| 200 trials after above | Contrast set = [100, 50, 25, 12.5, 6.25, 0] |
| 200 trials after above | Contrast set = [100, 25, 12.5, 6.25, 0] |
| 200 trials completed in previous session and reward volume > 1.5 µL | Decrease reward by 0.1 µL |
| Animal weight/25 > reward vol/1000 and reward volume < 3 µL | Next session increase reward by 0.1 µL |

**D**

| Proficiency level | Outcome |
|---|---|
| 'Basic task proficiency' - Trained 1a/1b<br>For each of the last three sessions:<br>>200/400 trials completed, and<br>>80%/90% correct on 100% contrast and all contrasts introduced and<br>For the last three sessions combined: psychometric absolute bias < 16/10 and psychometric threshold < 19/20 and psychometric lapse rates < 0.2/0.1<br>For 1b only: median reaction time at 0% contrast < 2 s | Training in the *basic task* achieved: mouse is ready to proceed to training in the *full task*. In some mice, we continued training in the *basic task* to obtain even higher performance.<br>(Guideline was to train mice for up to 40 days at Level 1, and drop mice from the study if they did not reach proficiency in this period). |
| 'Full task proficiency'<br>For each of the last three sessions:<br>>400 trials completed, and<br>>90% correct on 100% contrast and<br>For the last three sessions combined: all four lapse rates (left and right, 20:80 and 80:20 blocks) <0.1 and bias[80:20] - bias[20:80]>5 and median RT on 0% contrast < 2 s | Training in the *full task* achieved: mouse is ready for neural recordings. |

