## [Decision Letter]

**Acceptance summary:**

This is a very important work reporting a standardization of a decision-making task in mice. The result demonstrates that well-trained mice can reach behavioral performance consistent between several laboratories. The resulting behavioral training procedure and equipment designs will have a great impact on future experiments in the field.

**Decision letter after peer review:**

Thank you for submitting your article "Standardized and reproducible measurement of decision-making in mice" for consideration by *eLife*. Your article has been reviewed by 3 peer reviewers, and the evaluation has been overseen by a Reviewing Editor and Michael Frank as the Senior Editor. The reviewers have opted to remain anonymous.

The reviewers have discussed the reviews with one another and the Reviewing Editor has drafted this decision to help you prepare a revised submission.

Summary

In this manuscript, the International Brain Laboratory, an international consortium for systems neuroscience, reports their development of a mouse 2-alternative forced choice behavioral paradigm that incorporates both a perceptual decision (visual detection of a grating stimulus on the left side or right side) and the effect of priors/expectation on that decision (disproportionate presentation of the grating on the left or right side in alternating blocks of trials). The goal of the project was to develop a standardized task to address the crisis of reproducibility in rodent behavioral studies, with the overarching contention that variance in experimental results could be due to behavioral variability arising from methodological differences between labs. To address this issue, the consortium labs collectively planned the task and standardized the apparatus, software and training regimen for the 2AFC task; they set unified benchmarks for performance at different timepoints; and they met frequently to update each other on progress and to address pitfalls.

The results were mostly similar across laboratories, only with significant differences in the learning rates, and behavioral metrics that were not explicitly trained (trial duration and number of trials). However, once the mice learned the task, the behavior was extremely similar. Indeed, using the results from all laboratories, a classifier could not identify better from which laboratory the results came from in comparison to the shuffled results. Although it appears that the remaining variability in learning rates and some other task parameters still present some limitations in their standardization, the authors present evidence supporting a successful standardization in a behavioral paradigm.

Overall, all three reviewers thought that this is an important report that warrants publication at *eLife*. One reviewer thought that this is a landmark paper, which reports a fruitful, admirable, and convincing effort to demonstrate it is possible to achieve reproducibility in both methods and results of a perceptual decision-making task, across different laboratories around the world. However, all the reviewers raised various concerns regarding the methodology as well as the validity of specific conclusions. Furthermore, two reviewers thought that it would be helpful to clarify the merit of standardization in neuroscience. We therefore would like to invite the authors to respond to these concerns and suggestions before making a final decision.

Essential revisions:

1. One of the main points of the manuscript is that standardization of equipment, training procedures, and behavioral results has been achieved across labs. We regard this as a great achievement. Some of the metrics demonstrating this are shown in Figure 3b,c,d,e. However, these metrics are shown only for successfully trained animals, and the definition of successful training is that the animal has reached a given threshold on these metrics. In other words, only animals that have a metric within a narrow range are chosen to be included in the plots, and therefore, to some extent it is not surprising that all the animals in the plot lie within a narrow range. Furthermore, to someone reading the manuscript quickly, they might get the impression that the tight similarity across labs in 3b,c,d,e was not a result of selecting mice for inclusion in the figure, but instead a more surprising outcome. The main point here is that it would help if the authors expanded on this point and emphasized further clarity on it, since it is so central to the manuscript. (For example, if the criteria for successful training were simply a threshold on % correct, would one find similarity across labs in the chosen metrics to the same degree as with the selection procedure currently used?)

Along this line, to increase the clarity, it would be helpful to add some of the key parameters of the behavioral experiment to the methods, which are currently likely included only in the appendix. For example, what are the criteria for a successful trial? (how far can the mouse overshoot for it to count as a correct choice, how large is the target zone, how long does the stimulus need to be held there, or does it simply need to move it across the center line). Furthermore, how and at which stage in the project were these criteria for successful trials and animal exclusions established – before or after collecting the data?

2. Out of 210 mice, at least 66 subjects did not learn the basic task, and another 42 mice did not learn the full task version. It would be valuable to know which steps made the learning of the task difficult for these subjects and propose alternatives that might improve the number of subjects that learns the task. Were the subjects having difficulties to stop licking? Were mice not learning how to turn the wheel?

3. Related to the above point, do the authors know that the standardized protocol actually reduced end-point variability? One reviewer expressed that any behavioral neuroscientist would expect that if labs were really careful and open in standardizing their methodology, they should obtain similar behavioral performance from animals. Or to restate, how would one have interpreted the alternative result, that the different labs had found different performance? The obvious assumption would be that one or more labs screwed up and didn't follow the standardized protocol closely enough. We understand that the manuscript is intended as a technical resource but there is something vaguely disquieting about a report for which one possible outcome, non-uniform mouse behavior among labs, is almost surely discarded. If the effort is testing a hypothesis such as standardization enables consistent results, it would have been better to have conducted the study with pre-registration of the experiments and hypotheses, with a test of an alternative hypothesis (less strict standardization is sufficient). While the findings of uniform behavior is satisfying, we would like to hear how the authors would address these concerns.

4. In addition to the difference in learning rates, there was a significant difference in the number of trials. This could be due to differences in the session duration, which was not fixed but an experimenter would decide when it was convenient to end it. Thus, it could be that subjects that had longer sessions had the opportunity to complete a higher number of trials, and then learn the task in fewer sessions. Is the number of days to reach the learning criterion less variable across labs and mice if you group mice into sets of mice with similar not-explicitly trained variables (e.g., similar session durations, or similar number of trials per session)?

5. In motivating the task design, the authors claim to have combined perceptual decision-making with an assay from value-based decision making specifically two-armed bandits in referring to their stimulus-probability manipulation. However, all the references that the authors cite and two-armed bandits in general, manipulate probability/magnitude of rewards and NOT stimuli. In fact, the stimulus probability manipulation falls squarely under the realm of perceptual tasks (such as Hanks et al., 2011). Authors should remove their claims about value-based decision-making and cite appropriate studies for this stimulus prior manipulation instead of the incorrect references to bandit tasks.

6. Two reviewers raised the concern that it is unclear whether standardization in neurobiological experiments really facilitates scientific developments in the field. It would be useful for the field if the authors add some speculation of why they think it will be important to use a standardized behavioral protocol (i.e. where they see the advantage and how it will accelerate our understanding of brain function). Currently, most of the arguments are of the form 'standardization for the sake of standardization' – one could argue that a standardized task has similar potential pitfalls as if everybody would work on mouse visual cortex to try to understand cortical function. This would be up to the authors but given that a paper like this is rather rare, it would inspiring if the authors did that.

7. The authors used the animals' accuracy on the highest contrast gratings as a measure of progress in learning the task. However, this information is not immediately available had to dig into the figure legends to realize that. It would be helpful to label the y-axis on those figures to indicate that accuracy was defined for the highest-contrast trials.

8. It would be very helpful to have supplementary figures similar to the panels in Figure 2, but for bias and sensitivity (starting from when bias and sensitivity can be measured, of course.)

9. The reward is a very important component of the task. However, important details about the reward delivery system and whether licking is being tracked (through voltage/LED sensor or through video?) are missing. Is the reward immediately delivered after the correct response is emitted, even if the mouse has not licked the spout yet? Or is reward delivery contingent with licking? The reward delivery system is not included in the set-up scheme (Figure 1).

10. Please define precision and recall variables in Equation 1.

11. Please add an explanation of why a Frame2TTL is necessary "LCD screen refresh times were captured with a Bpod Frame2TTL (Sanworks)". Could it be that it allows a higher temporal precision track of when the stimuli are being delivered?

[Editors' note: further revisions were suggested prior to acceptance, as described below.]

Thank you for resubmitting your work entitled "Standardized and reproducible measurement of decision-making in mice" for further consideration by *eLife*. Your revised article has been evaluated by Michael Frank (Senior Editor) and a Reviewing Editor.

In this manuscript, the International Brain Laboratory, an international consortium for systems neuroscience, reports their development of a mouse 2-alternative forced choice behavioral paradigm that incorporates both a perceptual decision and the effect of priors/expectation on that decision. The authors find that the behavioral performance of the mice was largely consistent across the laboratories although there was some variability in the learning rate. This study provides the neuroscience community with valuable information regarding their efforts to achieve consistent behaviors as well as detailed protocols.

The manuscript has been improved and all reviewers agreed that the manuscript warrants publication at *eLife*. However, there are several remaining issues that the reviewers raised. The reviewers therefore would like to ask the authors to address these points before finalizing the manuscript.

Essential Revisions:

The reviewers thought that the manuscript has been greatly improved. While Reviewers 1 and 2 thought that the authors have fully addressed the previous concerns, Reviewer 2 asks several clarifications to which the authors should be able to address by revising the text. Reviewer 3 remains concerned that the authors' conclusions are still based on animals selected based on performance although the authors have now provided less selective criterion. Please revise the manuscript to address these issues before finalizing the manuscript.

*Reviewer #1:*

The authors have adequately addressed my initial concerns.

*Reviewer #2:*

We feel the paper could be published as is. We have a few final minor comments that could be straightforwardly addressed in writing.

1. The authors have corrected their description of the prior manipulation in the main text, clarifying that what they study in the manuscript is not what is usually studied in value-based decision-making tasks. However, they have not modified the abstract. The claim in the abstract about combining "established assays of perceptual and value-based decision-making" is not really correct. Please update this.

2. Overall, we believe the authors may have missed a good opportunity. Two out of the three reviewers felt the manuscript did not sufficiently articulate the value of the approach. A few lines were added to the discussion, but we feel more could have been done.

3. It would be helpful to readers if the authors comment on why this study and Roy 2021 (which analyzes the very same dataset) reach different conclusions about the evolution of contrast threshold over training. Roy 2021 claims that the contrast threshold decreases (i.e. sensory weights increases) over trials but the authors here report that it doesn't change much during training. (Lines 187-190).

4. The authors did explain why some subjects did not complete level 1, which include death, illness, or technical reasons. However, some subjects struggled with learning the first stage of the task (high bias, low performance, slow progress; n = 26 or 32), and another 42 mice did not learn the full task version. It would be desirable that the authors identify steps during training that hinder learning and which modifications would be made to tackle this issue. Maybe identifying the phase of the training when the learning curve of this slow- or non-learners diverges from proficient-subjects would allow further insight.

5. "All the papers on the subject told us that we should expect failure to replicate behavioral results across laboratories." Most of the papers cited did not report issues in replicating highly-stereotyped operant (i.e., turn left-right, press lever, lick) rodent behavior. If the authors have references describing highly-stereotyped operant rodent behavior that, used by different laboratories, turned out in discrepant behavioral results, these would be great to add.

6. Please include a section in Methods about how contrast threshold, bias etc. as reported in Figure 1-4 were inferred from choices.

*Reviewer #3:*

It appears there is a key piece of information missing in the authors responses describing their new analysis to address the concern of circularity in their conclusions (only including animals that learned to compare learning across laboratories). I can't find any information on how many mice were included in this less restrictive analysis (i may be missing this – but then this should be made more prominent). Why do the authors not include all mice for all analysis? Would that not be the point of standardizing behavior?

Overall, I do not feel the manuscript has improved much. Happy to provide a more thorough review, but the question above would seem central to answer first. Also, if this is the authors idea of a vision statement (or the reason for doing this in the first place) "Now that we have developed this task and established its reproducibility across laboratories, we are using it together with neural recordings, which are performed in different laboratories and combined into a single large data set. Other laboratories that adopt this task for studies of neural function will then be able to rely on this large neural dataset to complement their more focused results. Moreover, they will be able to compare each other's results knowing that any difference between them is unlikely to be due to differences in behavior." We, as a field, are in dire straits.

---

## [Author Response]

Essential revisions:1. One of the main points of the manuscript is that standardization of equipment, training procedures, and behavioral results has been achieved across labs. We regard this as a great achievement. Some of the metrics demonstrating this are shown in Figure 3b,c,d,e. However, these metrics are shown only for successfully trained animals, and the definition of successful training is that the animal has reached a given threshold on these metrics. In other words, only animals that have a metric within a narrow range are chosen to be included in the plots, and therefore, to some extent it is not surprising that all the animals in the plot lie within a narrow range. Furthermore, to someone reading the manuscript quickly, they might get the impression that the tight similarity across labs in 3b,c,d,e was not a result of selecting mice for inclusion in the figure, but instead a more surprising outcome. The main point here is that it would help if the authors expanded on this point and emphasized further clarity on it, since it is so central to the manuscript. (For example, if the criteria for successful training were simply a threshold on % correct, would one find similarity across labs in the chosen metrics to the same degree as with the selection procedure currently used?)

We agree that this is an important point, and our paper already contained an analysis that addresses it: we looked at performance on sessions that followed proficiency in the basic task (Figure 3 – supplement 5). Following the reviewers’ suggestions we added an additional analysis: instead of setting criteria on all three measures of behavior (% correct at high contrast, response bias, and contrast threshold), we set a criterion only on the first measure, putting the threshold at 80% correct. The sessions that led to this looser criterion thus were generally earlier, and no later, than the 3 sessions in Figure 3. The results are shown in a new figure (Figure 3 – supplement 4). As explained in Results, this new analysis gave similar results as the previous one: we found no significant differences in bias and sensory threshold across laboratories. Indeed, similar to the result obtained with the original training criteria, a decoder failed to identify the origin of a mouse based on its psychometric performance. This new analysis, therefore, confirms our results.

Along this line, to increase the clarity, it would be helpful to add some of the key parameters of the behavioral experiment to the methods, which are currently likely included only in the appendix. For example, what are the criteria for a successful trial? (how far can the mouse overshoot for it to count as a correct choice, how large is the target zone, how long does the stimulus need to be held there, or does it simply need to move it across the center line).

We thank the reviewers for this suggestion. We adopted our definition of a successfully registered response before we started data collection. We based it on the methods of Burgess et al. (2017). We now describe it in Methods.

Furthermore, how and at which stage in the project were these criteria for successful trials and animal exclusions established – before or after collecting the data?

We thank the reviewers for this suggestion. We now explain this timeline in a new paragraph in Methods.

2. Out of 210 mice, at least 66 subjects did not learn the basic task, and another 42 mice did not learn the full task version. It would be valuable to know which steps made the learning of the task difficult for these subjects and propose alternatives that might improve the number of subjects that learns the task. Were the subjects having difficulties to stop licking? Were mice not learning how to turn the wheel?

We agree that this is useful information, and we have done our best to answer it. However, our database contains only brief explanations for why a mouse was dropped, so we cannot do a thorough study of this aspect. We now document this in Methods.

3. Related to the above point, do the authors know that the standardized protocol actually reduced end-point variability? One reviewer expressed that any behavioral neuroscientist would expect that if labs were really careful and open in standardizing their methodology, they should obtain similar behavioral performance from animals. Or to restate, how would one have interpreted the alternative result, that the different labs had found different performance? The obvious assumption would be that one or more labs screwed up and didn't follow the standardized protocol closely enough.

As we explain in the paper, before we started this work all the papers on the subject told us that we should expect failure to replicate behavioral results across laboratories. We were thus uncertain of success, and particularly uncertain given that we have so many laboratories distributed across the world. The results indicated that there are some things that we can fully reproduce across labs (e.g. key aspects of the psychometric curves once mice have learned the task) and others that we can’t (learning rates). So, some of our results are positive but some are negative.

As for the previous studies that had failed to replicate mouse behavior across labs, we trust that their authors were indeed “really careful” (to quote the review). We don’t think the failure to replicate is due to shortcomings in the way the experiments were run. Rather, we believe that they chose to measure aspects of behavior that are not easy to replicate (e.g. a mouse’s willingness to traverse an exposed raised corridor). We devote the 4th paragraph of Discussion to this issue.

We understand that the manuscript is intended as a technical resource but there is something vaguely disquieting about a report for which one possible outcome, non-uniform mouse behavior among labs, is almost surely discarded. If the effort is testing a hypothesis such as standardization enables consistent results, it would have been better to have conducted the study with pre-registration of the experiments and hypotheses, with a test of an alternative hypothesis (less strict standardization is sufficient). While the findings of uniform behavior is satisfying, we would like to hear how the authors would address these concerns.

We could not have discarded our results if they had been negative because we are part of a large open-science effort that was known to the public (we published our manifesto in *Neuron* in 2018) and that involved a large number of stakeholders: multiple funders, multiple labs, multiple students and postdocs. Whatever the result, we would have had to publish it, both for reasons of scientific fairness and of duties to the funders and more practically to recognize the work done by students and postdocs.

A similar consideration, indeed, applies to previous efforts to replicate behavioral results across labs. Indeed, there are many papers on failure to replicate mouse behavior across labs, and zero papers about success. Whenever multiple labs get together and try to establish methods that are shared and reproducible, one can expect a paper reporting the results.

This is arguably another advantage of team science, it keeps things honest, preventing the burying of negative results. In this sense, it is a form of pre-registration. At a larger scale, this happens routinely in high-energy physics: it is widely known that a particle accelerator is being run on a certain experiment, so we can be sure that the results of the experiment will be released whether they are positive or negative.

As for the alternative hypothesis, it is possible that a somewhat lower degree of standardization would have sufficed, but none of our standardized procedures would be particularly onerous on a lab that would like to adopt our methods, so we do not see this as a pressing matter.

4. In addition to the difference in learning rates, there was a significant difference in the number of trials. This could be due to differences in the session duration, which was not fixed but an experimenter would decide when it was convenient to end it. Thus, it could be that subjects that had longer sessions had the opportunity to complete a higher number of trials, and then learn the task in fewer sessions. Is the number of days to reach the learning criterion less variable across labs and mice if you group mice into sets of mice with similar not-explicitly trained variables (e.g., similar session durations, or similar number of trials per session)?

We see the reviewer’s point: mice that performed longer sessions might learn in fewer days simply because they are doing more trials per day. To address this possibility we have made a new Supplementary figure (Figure 2 – supplement 1) where we plotted performance as a function of trial number rather than day of training. The results remained the same. This is now discussed in a new paragraph in Results. We also added results about learning as a function of trials in Figure 2 – supplement 2.

5. In motivating the task design, the authors claim to have combined perceptual decision-making with an assay from value-based decision making specifically two-armed bandits in referring to their stimulus-probability manipulation. However, all the references that the authors cite and two-armed bandits in general, manipulate probability/magnitude of rewards and NOT stimuli. In fact, the stimulus probability manipulation falls squarely under the realm of perceptual tasks (such as Hanks et al., 2011). Authors should remove their claims about value-based decision-making and cite appropriate studies for this stimulus prior manipulation instead of the incorrect references to bandit tasks.

We thank the reviewers for this suggestion and for alerting us to the study by Hanks and colleagues, which we now cite. We have rewritten that paragraph in Introduction, to make more links to studies of perceptual decision making, and to clarify that the analogy with two-armed bandit tasks applies only to the case where there are no stimuli.

6. Two reviewers raised the concern that it is unclear whether standardization in neurobiological experiments really facilitates scientific developments in the field. It would be useful for the field if the authors add some speculation of why they think it will be important to use a standardized behavioral protocol (i.e. where they see the advantage and how it will accelerate our understanding of brain function). Currently, most of the arguments are of the form 'standardization for the sake of standardization' – one could argue that a standardized task has similar potential pitfalls as if everybody would work on mouse visual cortex to try to understand cortical function. This would be up to the authors but given that a paper like this is rather rare, it would inspiring if the authors did that.

The current situation for labs where animals perform tasks is that every lab does a slightly different task and records from different neurons. If results disagree, one does not know if that’s because the task is different or because the neurons are different. We are trying to remedy this situation, providing a standard that could be useful to many laboratories. Such standards are sorely needed whenever there is an effort that requires more than one laboratory.

We think this comes across in the paper but to be sure it does, we have added sentences in the last paragraph of Discussion.

7. The authors used the animals' accuracy on the highest contrast gratings as a measure of progress in learning the task. However, this information is not immediately available had to dig into the figure legends to realize that. It would be helpful to label the y-axis on those figures to indicate that accuracy was defined for the highest-contrast trials.

We changed Figure 1e to clarify that it’s performance on easy trials, and we added a clarification in the legend.

8. It would be very helpful to have supplementary figures similar to the panels in Figure 2, but for bias and sensitivity (starting from when bias and sensitivity can be measured, of course.)

We agree, and we have added 16 panels to Figure 2 to show this. New text in Results describes those data.

9. The reward is a very important component of the task. However, important details about the reward delivery system and whether licking is being tracked (through voltage/LED sensor or through video?) are missing. Is the reward immediately delivered after the correct response is emitted, even if the mouse has not licked the spout yet? Or is reward delivery contingent with licking? The reward delivery system is not included in the set-up scheme (Figure 1).

We edited the Methods to explain that reward was not contingent on licking, and therefore licking was not tracked online. We also added a sentence mentioning the video tracking of the animal during the experiment. We replaced the schematic of the rig in Figure 1c, to include the reward spout.

10. Please define precision and recall variables in Equation 1.

We thank the reviewer for this suggestion. We have now simplified the language and the description, and we now have dropped the words “F1 score”, “precision”, and “recall”, opting for the simpler “decoding accuracy”. The equation that describes accuracy ( equation 1) is simpler but it is mathematically equivalent to our previous equation defining F1.

11. Please add an explanation of why a Frame2TTL is necessary "LCD screen refresh times were captured with a Bpod Frame2TTL (Sanworks)". Could it be that it allows a higher temporal precision track of when the stimuli are being delivered?

We thank the reviewers for this comment, which led us to explain this rationale in Methods.

[Editors' note: further revisions were suggested prior to acceptance, as described below.]

Essential Revisions:The reviewers thought that the manuscript has been greatly improved. While Reviewers 1 and 2 thought that the authors have fully addressed the previous concerns, Reviewer 2 asks several clarifications to which the authors should be able to address by revising the text. Reviewer 3 remains concerned that the authors' conclusions are still based on animals selected based on performance although the authors have now provided less selective criterion. Please revise the manuscript to address these issues before finalizing the manuscript.Reviewer #2:We feel the paper could be published as is. We have a few final minor comments that could be straightforwardly addressed in writing.1. The authors have corrected their description of the prior manipulation in the main text, clarifying that what they study in the manuscript is not what is usually studied in value-based decision-making tasks. However, they have not modified the abstract. The claim in the abstract about combining "established assays of perceptual and value-based decision-making" is not really correct. Please update this.

Thank you, we have now fixed this. We replaced “designed” with “adopted”, and we removed “combines”. The resulting sentence reads: “We adopted a task for head-fixed mice that assays perceptual and value-based decision making.” Indeed, the task assays perceptual decision making because performing it well requires sensory processing; it assays value-based decision making because at zero contrast it is exactly the same as a two-armed bandit task. In Introduction we clarify that the task was developed by people who study perceptual decision making, as the reviewers have pointed out.

2. Overall, we believe the authors may have missed a good opportunity. Two out of the three reviewers felt the manuscript did not sufficiently articulate the value of the approach. A few lines were added to the discussion, but we feel more could have been done.

We thank the reviewers for pushing us in this direction.

We have made some edits to the Introduction, where the first three paragraphs explain that (1) Progress in science depends on reproducibility, but neuroscience faces a crisis of reproducibility; (2) Reproducibility has been a particular concern for measurements of mouse behavior; (3) A difficulty in reproducing mouse behavior across laboratories would be problematic for the increasing number of studies that investigate decision making in the mouse. Also in Introduction, we have strengthened the point about the value of Open Science approaches.

We have also added a paragraph in Discussion: “A reproducible behavioral task can be invaluable to establish the neural basis of behavior. If different studies use the same task, they can directly compare their findings. There are indeed illustrious examples of behavioral tasks that serve this role. For studying decision-making in primates, these include the tactile flutter comparison task […] and the random dots visual discrimination task […]. Both tasks have been used in multiple studies to record from different brain regions while enabling a meaningful comparison of the results. Conversely, without a standardized behavioral task we face the common situation where different laboratories record from different neurons in different regions in different tasks, likely drawing different conclusions and likely not sharing their data. In that situation it is not possible to establish which factors determine the different conclusions and come to a collective understanding.”

3. It would be helpful to readers if the authors comment on why this study and Roy 2021 (which analyzes the very same dataset) reach different conclusions about the evolution of contrast threshold over training. Roy 2021 claims that the contrast threshold decreases (i.e. sensory weights increases) over trials but the authors here report that it doesn't change much during training. (Lines 187-190).

We thank the reviewer for bringing this up. We have added some quantification, which shows a modest decrease in threshold during the first days of training. We also added a summary across laboratories that illustrates this effect more clearly (black curve in Figure 2e-f).

However these effects cannot be directly compared to those examined by Roy et al. (2021), for two reasons: (1) Roy et al. use a method that can look at all sessions, even the earliest ones where there are only two (high) contrasts per side, whereas our analysis requires having more contrasts so that we can fit a psychometric curve; (2) The model by Roy et al. involves no equivalent of a lapse rate. Mice can improve the performance in the task by both decreasing their lapse rate and/or putting more weight on the sensory evidence. Our psychometric model parametrizes both of these features separately; (3) The paper by Roy et al. (2021) used some mice as examples, without attempting to describe a vast dataset as we do here.

4. The authors did explain why some subjects did not complete level 1, which include death, illness, or technical reasons. However, some subjects struggled with learning the first stage of the task (high bias, low performance, slow progress; n = 26 or 32), and another 42 mice did not learn the full task version. It would be desirable that the authors identify steps during training that hinder learning and which modifications would be made to tackle this issue. Maybe identifying the phase of the training when the learning curve of this slow- or non-learners diverges from proficient-subjects would allow further insight.

We thank the reviewer for this question. This is an area of ongoing research in our collaboration. To summarize the main result that we have so far, we have added the following paragraph to Results: “To some extent, a mouse’s performance in the first 5 sessions predicted how long it would take the mouse to become proficient. A Random Forests decoder applied to change in performance (% correct with easy, high-contrast stimuli) in the first 5 sessions was able to predict whether a mouse would end up in the bottom quartile of learning speed (the slowest learners) with accuracy of 53% (where chance is 25%). Conversely, the chance of misclassifying a fast-learning, top quartile mouse with a slow-learning, bottom quartile mouse, was only 7%.”

5. "All the papers on the subject told us that we should expect failure to replicate behavioral results across laboratories." Most of the papers cited did not report issues in replicating highly-stereotyped operant (i.e., turn left-right, press lever, lick) rodent behavior. If the authors have references describing highly-stereotyped operant rodent behavior that, used by different laboratories, turned out in discrepant behavioral results, these would be great to add.

We agree, and we do already point this out: “Previous failures to reproduce mouse behavior across laboratories typically arose in studies of unconstrained behavior such as responses to pain or stress.” Following the reviewer’s comment we have now added a sentence: “Operant behaviors may be inherently more reproducible”.

6. Please include a section in Methods about how contrast threshold, bias etc. as reported in Figure 1-4 were inferred from choices.

Thank you for this suggestion. Those methods were previously described in the Appendix 2 and we have now moved them to Methods.

Reviewer #3:It appears there is a key piece of information missing in the authors responses describing their new analysis to address the concern of circularity in their conclusions (only including animals that learned to compare learning across laboratories). I can't find any information on how many mice were included in this less restrictive analysis (i may be missing this – but then this should be made more prominent). Why do the authors not include all mice for all analysis? Would that not be the point of standardizing behavior?

We assume that this comment refers to the new analysis that we performed in the last round of review, where instead of setting criteria on all three measures of behavior (% correct at high contrast, response bias, and contrast threshold), we set a criterion only on the first measure, putting the threshold at 80% correct. The sessions that led to this looser criterion thus were generally earlier, and no later, than the 3 sessions in Figure 3. This analysis is shown in Figure 3 – supplement 4. Of course we have run this analysis on all mice that passed that criterion (80% correct on easy trial), with no other data selection. We have now updated the figure legend to give the exact number: n = 150.

Overall, I do not feel the manuscript has improved much. Happy to provide a more thorough review, but the question above would seem central to answer first. Also, if this is the authors idea of a vision statement (or the reason for doing this in the first place) "Now that we have developed this task and established its reproducibility across laboratories, we are using it together with neural recordings, which are performed in different laboratories and combined into a single large data set. Other laboratories that adopt this task for studies of neural function will then be able to rely on this large neural dataset to complement their more focused results. Moreover, they will be able to compare each other's results knowing that any difference between them is unlikely to be due to differences in behavior." We, as a field, are in dire straits.

The reviewer does not give us guidance as to what would constitute an appropriate “vision statement” in our Discussion. We are thus not sure what is being requested. In hopes of hitting the mark, we have added a new paragraph: “A reproducible behavioral task can be invaluable to establish the neural basis of behavior. If different studies use the same task, they can directly compare their findings. There are indeed illustrious examples of behavioral tasks that serve this role. For studying decision-making in primates, these include the tactile flutter comparison task […] and the random dots visual discrimination task […]. Both tasks have been used in multiple studies to record from different brain regions while enabling a meaningful comparison of the results. Conversely, without a standardized behavioral task we face the common situation where different laboratories record from different neurons in different regions in different tasks, likely drawing different conclusions and likely not sharing their data. In that situation it is not possible to establish which factors determine the different conclusions and come to a collective understanding.”